# Motility and microtubule depolymerization mechanisms of the Kinesin-8 motor, KIF19A

Doudou Wang[1,2], Ryo Nitta[3], Manatsu Morikawa[1,2], Hiroaki Yajima[1,2], Shigeyuki Inoue[1,2], Hideki Shigematsu[3], Masahide Kikkawa[1], Nobutaka Hirokawa[1,2,4]*

[1]Department of Cell Biology and Anatomy, Graduate School of Medicine, The University of Tokyo, Tokyo, Japan; [2]Department of Molecular Structure and Dynamics, Graduate School of Medicine, The University of Tokyo, Tokyo, Japan; [3]RIKEN Center for Life Science Technologies, Yokohama, Japan; [4]Center of Excellence in Genomic Medicine Research, King Abdulaziz University, Jeddah, Saudi Arabia

**Abstract** The kinesin-8 motor, KIF19A, accumulates at cilia tips and controls cilium length. Defective KIF19A leads to hydrocephalus and female infertility because of abnormally elongated cilia. Uniquely among kinesins, KIF19A possesses the dual functions of motility along ciliary microtubules and depolymerization of microtubules. To elucidate the molecular mechanisms of these functions we solved the crystal structure of its motor domain and determined its cryo-electron microscopy structure complexed with a microtubule. The features of KIF19A that enable its dual function are clustered on its microtubule-binding side. Unexpectedly, a destabilized switch II coordinates with a destabilized L8 to enable KIF19A to adjust to both straight and curved microtubule protofilaments. The basic clusters of L2 and L12 tether the microtubule. The long L2 with a characteristic acidic-hydrophobic-basic sequence effectively stabilizes the curved conformation of microtubule ends. Hence, KIF19A utilizes multiple strategies to accomplish the dual functions of motility and microtubule depolymerization by ATP hydrolysis.

*For correspondence: hirokawa@
m.u-tokyo.ac.jp

**Competing interests:** The authors declare that no competing interests exist.

## Introduction

Kinesin superfamily proteins (KIFs) are microtubule-based molecular motors that hydrolyze ATP to provide energy to support various cellular functions, such as intracellular transport and the regulation of microtubule (MT) dynamics (*Hirokawa et al., 2009*; *Miki et al., 2005*). Most KIFs, including kinesin-1–kinesin-12 and kinesin-14 subfamily members, actively move along the MT toward either the plus or minus end to transport cellular cargoes, such as protein complexes, vesicles and mRNAs (*Hirokawa et al., 2009*). In contrast, kinesin-13 proteins, such as KIF2A and KIF2C, are not actively motile along MTs but depolymerize MTs from both ends to control MT dynamics (*Desai et al., 1999*; *Homma et al., 2003*; *Ogawa et al., 2004*). Kinesin-13 proteins reach the ends of MTs by a passive one-dimensional random diffusion (*Helenius et al., 2006*) or with the assistance of a MT plus end-tracking protein (*Honnappa et al., 2009*). KIF19A, a kinesin-8 sub-family member, is unique in that it possesses both functions, MT-based active motility toward the plus-end and MT depolymerizing activity.

Most kinesin-8 motor proteins play critical roles during the cell division process, for example in spindle length regulation (*Weaver et al., 2011*; *Su et al., 2013*) or in the control of mitotic chromosome alignment (*Kline-Smith and Walczak, 2004*; *Stumpff et al., 2008*). The most extensively

**eLife digest** The cells that line the airways and other passages in the body have hair-like structures called cilia on their surface. Maintaining the cilia at an appropriate length is key to allowing fluid to flow efficiently in these passages. A protein called tubulin forms scaffolds known as microtubules that give each cilium its shape and allow it to change length.

Motor proteins are also found in cilia, and travel along the microtubules to transport substances. One of these microtubule-based motors, referred to as KIF19A, accumulates at the tip of cilia and controls their length. It does so by combining two actions: it moves along the microtubule to the tip of the cilium, and then removes tubulin molecules from the end. Microtubules are straight along their length and curved at the end, and it is thought that kinesin recognizes both of these shapes in order to carry out these roles. A single region of the KIF19A protein appears to be able to accomplish both roles, but the molecular changes that the protein undergoes to do so are not known.

Wang et al. have now investigated these changes by determining the structure of the motor domain of KIF19A from mice using two experimental approaches: X-ray crystallography and cryo-electron microscopy. These structures showed that the specific structural features responsible for the protein's dual roles are indeed clustered on the side of the protein that binds to the microtubule. Wang et al. also identified the regions that make KIF19A flexible enough to fit this interface with both straight and curved microtubules.

Next, Wang et al. found that other regions of KIF19A stop it detaching from the microtubule and allow it to stabilize the curved shape of microtubule ends; this stimulates the microtubule to disassemble, or "depolymerize". The findings show that KIF19A uses multiple strategies to enable it to carry out its roles. To understand better how KIF19A depolymerizes the microtubule, a more detailed structure of KIF19A together with tubulin will be needed. Structural studies of KIF19A in cilia will also be useful to understand how the protein controls the length of microtubules.

studied kinesin-8, Kip3p (budding yeast kinesin-8; a homologue of mammalian KIF18), requires its C-terminal tail for MT-binding to prevent its detachment from the MT lattice (*Mayr et al., 2011*; *Stumpff et al., 2011*; *Su et al., 2011*; *Weaver et al., 2011*). Kip3p is highly processive (>5 µm) and multiple Kip3ps act cooperatively to mediate length-dependent MT depolymerization. One possible model for a length-dependent action is proposed to involve the incoming Kip3p bumping off the paused motor at the MT plus-end (*Gupta et al., 2006*; *Varga et al., 2006*, *2009*). Structural studies of the motor domain of the human kinesin-8, KIF18A, showed a bent conformation of the $\alpha 4$ relay helix and important loops, indicating the structural basis of the multi-tasking kinesin-8 motor (*Peters et al., 2010*). However, the molecular mechanisms of the dual-functions of kinesin-8 proteins remain to be determined.

We recently reported that KIF19A functions as a MT-depolymerizing kinesin in the control of cilium length (*Niwa et al., 2012*). $Kif19a^{-/-}$ mice displayed hydrocephalus and female infertility phenotypes due to abnormally elongated cilia that cannot generate proper fluid flow. We also reported that, unlike KIF18A, a KIF19A dimer without the tail domain depolymerizes MTs mainly from the plus-end. Therefore, KIF19A possesses the key structural elements for the dual functions of the catalytic motor domain. Thus, to elucidate the molecular mechanism of the dual KIF19A functions, we performed crystal structure analysis of the mouse KIF19A motor domain as well as cryo-electron microscopy (cryo-EM) reconstruction of the KIF19A motor domain complexed with a MT. In combination with a structure-based mutagenesis analysis, the functional anatomy of KIF19A is reported. In the catalytic core of KIF19A, the KIF19A-specific feature of adopting two different interfaces for MTs and tubulins is utilized to achieve the dual functions.

## Results

### KIF19A monomer is a dual function motor

We previously reported that dimeric KIF19A-379 has dual activities: MT-based motility toward the plus-end and MT-depolymerizing activity mainly from the plus-end (*Niwa et al., 2012*). To clarify which region is responsible for these dual functions, we made the monomeric construct KIF19A-353 (353WT) and assessed its motility and MT-depolymerizing activities. 353WT includes the motor domain followed by the neck-linker, but does not include the neck coiled coil, which is required for the dimerization of KIF19A (*Figure 1A*). We first performed the *in vitro* MT gliding assay, in which tetramethylrhodamine (TMR)-labeled and polarity-marked MTs were used to show the tracking

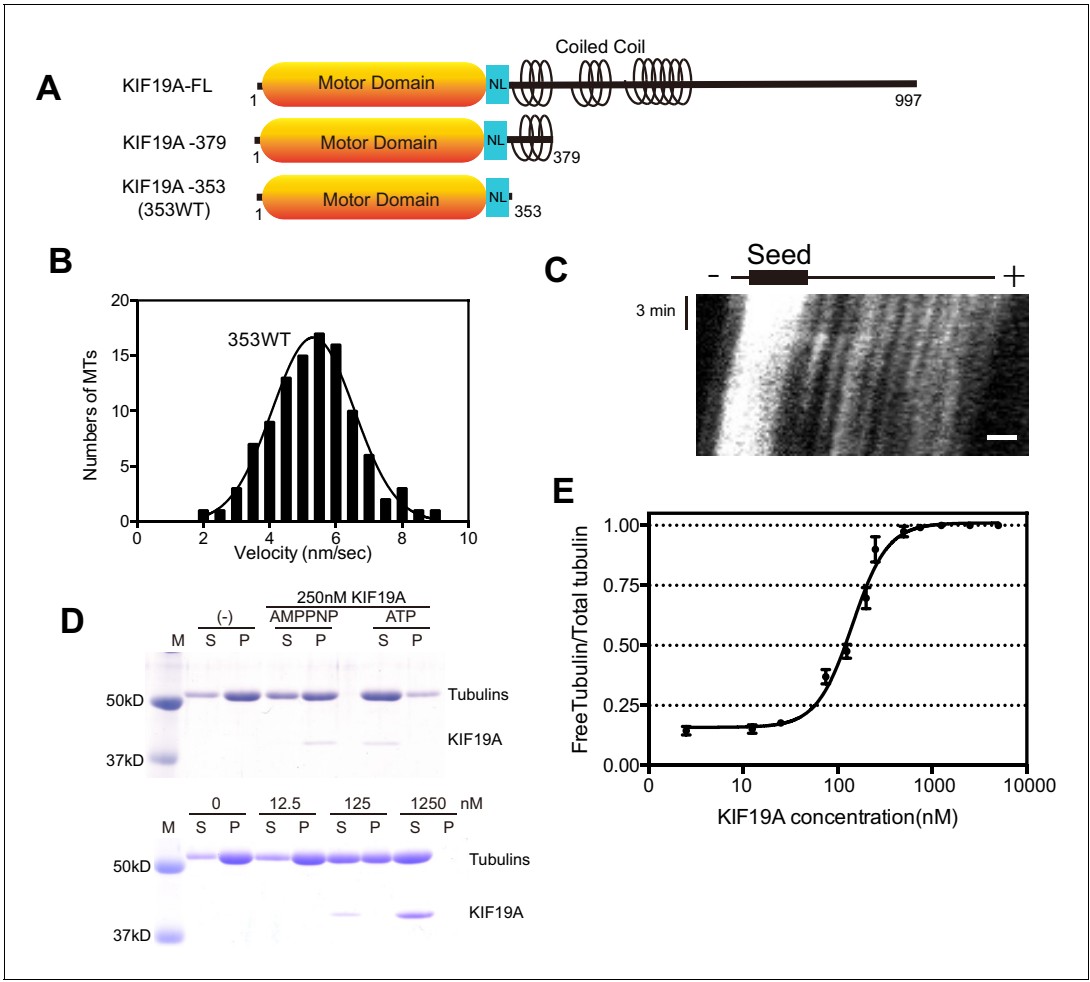

**Figure 1.** Characteristics of the dual function mouse KIF19A motor domain. (**A**) Schematic of mouse KIF19A motor domain constructs. The KIF19A monomer KIF19A-353 (referred to as 353WT) was used in this study. NL, neck-linker. (**B**) MT gliding assays of 353WT on taxol-stabilized MTs. Data are presented as the mean ± SEM, n = 105 MTs. (**C**) Kymograph showing movement of 353WT along MTs as imaged by TIRF microscopy. Scale bars, 2 μM (horizontal) and 3 min (vertical). The average MT length is 8.39 ± 2.71 μM. (**D**) 1.5 μM GMPCPP-stabilized MTs were incubated with 250 nM 353WT in the presence of 5 mM ATP or AMP-PNP (Top) for 15 min. AMP-PNP was used as a control. 1.5 μM GMPCPP-stabilized MTs were incubated with different concentrations of 353WT in the presence of 5 mM ATP for 15 min (Bottom). The microtubules were pelleted by centrifugation to separate the free tubulin (S) and MT pellets (P). SDS-PAGE and Coomassie brilliant blue staining were used to confirm the ratio of free tubulin and MTs. Representative data from three independent sample preparations are shown. (**E**) Dose-response MT depolymerization curve for different concentrations of 353WT. Data are presented as the mean ± SD. The mean $EC_{50}$ values of 353WT was 142 ± 2 nM. The data from three independent experiments were analysed.

The following source data is available for figure 1:

**Source data 1.** The data and analysis for 353WT.

direction. The strongly-labeled MT minus-ends lead the MT gliding, suggesting that the monomeric 353WT moves toward the plus-end (*Video 1*). MT gliding velocity was 5.3 ± 1.2 nm/s (n = 105 MTs from three independent preparations, mean ± SD, *Figure 1B and C*), which was slower than that of dimeric KIF19A-379 (21 ± 3 nm/s) (*Niwa et al., 2012*). An in vitro MT depolymerizing assay was also performed for KIF19A-353 (*Desai et al., 1999*). GMPCPP-MTs were dose-dependently depolymerized by 353WT (*Figure 1D*). The half-maximal effective concentration for MT depolymerization ($EC_{50}$) was 142 ± 2 nM, which was approximately half that of KIF19A-379 (253 nM) (*Figure 1E*). Considering that one of two motor domains will reach the plus-end of the MT, $EC_{50}$ values of one catalytic unit for depolymerizing MTs might be similar between monomeric 353WT and dimeric KIF19A-379. Either way, these in vitro experiments collectively indicate that the KIF19A monomer construct, 353WT, is a dual function motor that moves along and depolymerizes MTs.

## Crystal structure of the mouse KIF19A motor domain in the ADP state

We investigated the molecular mechanism of the dual functions of the KIF19A 353WT monomeric motor by solving its crystal structure at 2.7 Å resolution (*Supplementary file 1*). At the initial stage of the 353WT structure determination process, the residues of the switch II helix α4 could not be determined because there was no corresponding density at the site where α4 is usually located at the center of the MT-binding interface (*Figure 2—figure supplement 1A*). Instead, we found an unmodeled helical-like density, which was more distant from the KIF19A catalytic core (*Figure 2—figure supplement 1B*). Long wavelength X-ray diffraction experiment was thus performed to investigate the property of this density (*Hendrickson and Teeter, 1981*; *Dauter et al., 1999*), because in the switch II helix α4, one cysteine residue exists. It successfully depicted the anomalous diffractions of sulfur or phosphorus atoms (*Figure 2—figure supplement 1C–E*). Among them, one strong anomalous signal was detected close to the center of the corresponding helical density (*Figure 2—figure supplement 1E*). When the Cys283 residue was assigned to this anomalous signal, all the residues in α4 and the following loop L12 were reasonably determined (*Figure 2—figure supplement 1F*). In this way, the complete tertiary structure of 353WT was finally determined with a reasonable decrease of the R-factor to 22.2% (*Supplementary file 1*).

The overall structure of the KIF19A 353WT motor domain shared a similar triangle-shape with other kinesins, which consists of a central β-sheet of eight strands, sandwiched between six α-helices, three on either side (*Figure 2A and B*) (*Sack et al., 1999*). In the 353WT atomic structure, $Mg^{2+}$-ADP was found embedded in the nucleotide-binding pocket. In comparison with previously solved motor domains of various KIFs, the remarkable features of the KIF19A motor domain are concentrated on its MT-binding side and include the long and wide L2 loop, the flexible L8 loop, the disordered α4 helix, and the short α6 helix (*Figure 2A and B*). Loop L2 is located at the minus-end side (rear side) of the MT-binding surface and is atypically long. Loop L8, which is the major MT-binding region at the plus-end side, is more retracted toward the catalytic core compared with that of kinesin-13 (*Figure 2C*). The switch II helix α4, which is considered the main contributor to MT-based kinesin motility, is more distant from the motor domain than those of other kinesins including kinesin-8 KIF18A (*Figure 2C and D*). In comparison to the other kinesins, the first portion of helix α5 is more destabilized in KIF19A (*Figure 2F*); therefore, the loop region between α4 and α5 (L12) is longer. This might allow the invasion of a neighboring molecule into the space between α4 and the motor domain in the crystal packing environment, resulting in the distant position of α4 (*Figure 2—figure supplement 1G*). Helix α6 serves as a base for the neck-linker element, which is important for both the regulation of ATPase activity and MT-based kinesin motility (*Case et al., 2000*). Its reduced length by more than one turn might be caused by the atypical

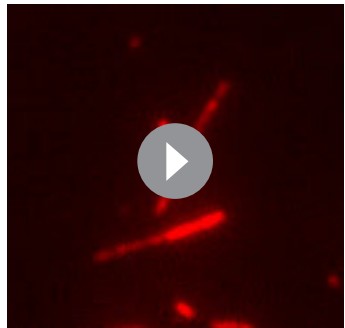

**Video 1.** Polarity-marked microtubules sliding on 353WT. 353WT was fixed on the coverslip. The strongly-labeled MT minus-ends lead the MT gliding, indicating that KIF19A motor proteins move toward the plus end. 10 seconds intervals, total tracking time 15 minutes.

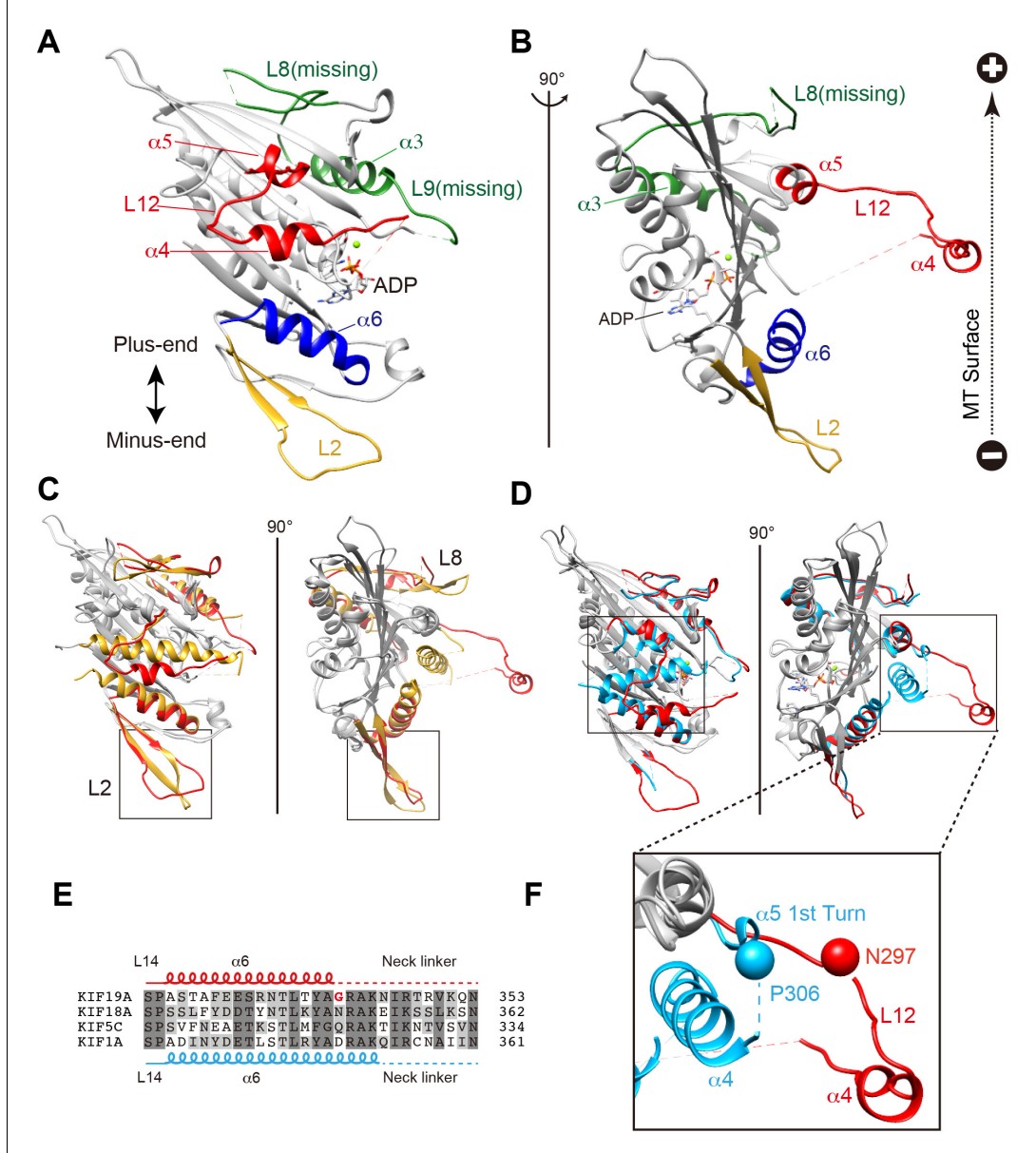

**Figure 2.** Crystal structure of the dual function mouse KIF19A motor domain. (A, B) Crystal structure of the KIF19A motor domain in the ADP-state seen from the MT binding side (A) and 90 degree rotation around the ordinate axis (B). L2 (yellow), L8-α3-L9 (green), L11-α4-L12-α5 (red) and α6 (blue) are depicted. The Mg$^{2+}$-ADP is shown as a ball and stick model. (C) Superposition of the KIF19A motor domain and KIF2C-ADP (PDB: 1V8J). The clusters with obviously different structures are colored red (KIF19A) and gold (KIF2C). L2 is shown in the square. (D) Superposition of KIF19A-ADP and KIF18A-ADP (PDB: 3LRE). The clusters with obviously different structures are colored red (KIF19A) and light blue (KIF18A). α4-L12-α5 is shown in the square. (E) Sequence alignment of α6 between KIF19A and other typical moving kinesin motors. For other kinesin motors see *Figure 2—figure supplement 2A*. (F) Zoom-in view of α4-L12-α5 of KIF19A and KIF18A motor domain in *Figure 2D*. N297 of KIF19A and P306 of KIF18A are shown as balls.

The following figure supplements are available for figure 2:

**Figure supplement 1.** The unmodeled helix determination by the long wavelength X-ray diffraction.

**Figure supplement 2.** Sequence alignment between representative kinesin members.

glycine mutation in helix α6, which is not seen in other typical plus end motors (G341, *Figure 2E* and *Figure 2—figure supplement 2A*). Glycine has high conformational flexibility and often tends to disrupt the α-helix, which might affect the conformation of the following neck-linker by terminating α6 prematurely.

## Basic and hydrophobic residues in L2 are crucial for MT depolymerizing activity

To identify structural elements contributing to the dual functions of KIF19A, we first investigated the loop L2. L2 of kinesin-13 is a key element contributing to MT depolymerization (*Ogawa et al., 2004*; *Shipley et al., 2004*). According to sequence alignments, KIF2C and KIF19A have L2 regions (β1b-L2-β1c) of comparable length in which the hydrophobic residue(s) around the tip is/are suggested to be surrounded by basic and acidic residue(s) (*Figure 3A*). KIF2C has a slender L2 with a KVD finger at its tip that is vital to MT depolymerization (*Figure 3B*) (*Ogawa et al., 2004*; *Shipley et al., 2004*). In contrast, KIF19A has a long, fan-shaped L2 that has a hydrophobic tip (I54-L55) sandwiched between acidic and basic clusters (*Figure 3B*). Despite the acidic-hydrophobic-basic order being conserved between KIF2C and KIF19A, their shapes are distinct from each other. To test the contribution of these residues to MT-depolymerizing activity, we introduced a series of mutations: an alanine mutant of the basic cluster (PC2A: R56A-H58A-R59A-R61A) and hydrophobic mutants (I54A, L55A, and IL2A: I54A-L55A). It should be noted that the alanine mutant of the acidic cluster was so unstable that we were unable to acquire reliable *in vitro* data. The acidic cluster is, however, expected to contribute to the depolymerization function.

*In vitro* MT depolymerization assays of L2 mutants were performed using a saturated concentration (250 nM) of 353WT (*Figure 3C*). PC2A, L55A and IL2A markedly impaired depolymerization, while I54A had little effect (*Figure 3C*). Different concentrations of 353WT and the L2 mutants that showed an effect (PC2A and L55A) were then incubated with GMPCPP-stabilized MTs to obtain $EC_{50}$ values for MT depolymerization. The mean $EC_{50}$ values of 353WT, PC2A and L55A in three independent experiments were $142 \pm 2$ nM, $4936 \pm 15$ nM and $409 \pm 4$ nM, respectively (*Figure 3D*). The dose-reaction curve of L55A was shifted to the right of 353WT (*Figure 3D*). For PC2A, even at the highest enzyme concentration used (5000 nM), approximately 50% of the wild-type depolymerization activity was achieved. We also observed MT depolymerization in the presence of 5 mM Mg-ATP by TIRF microscopy (*Figure 3E*). The depolymerization was mainly observed at the MT plus-ends and the speeds were $10.9 \pm 2.0$ nm/s for 353WT, $2.5 \pm 0.5$ nm/s for L55A, and $0.7 \pm 0.2$ nm/s for PC2A. The depolymerization speed of PC2A was the lowest among them, and only a little higher than that achieved by the negative control ($0.2 \pm 0.02$ nm/s) (*Figure 3F*).

We then tested the effect of introducing KIF19A L2 into KIF18A (another kinesin-8 member) and KIF5C (a typical plus-end directed motor) (*Figure 3G*). Consistent with a previous report, the KIF18A monomer had little MT depolymerization activity (*Weaver et al., 2011*). The substitution of KIF19A L2 mildly increased the depolymerization activity of KIF18A. Note that a fraction of KIF18A moved to the soluble fraction via interaction with depolymerized tubulin-dimers. In contrast, the L2 of KIF19A did not make KIF5C a MT depolymerase. These data collectively suggested that the basic cluster of L2, as well as the Leu 55 at the tip of L2, play crucial roles in depolymerization from MT ends, although the simple swapping of L2 is not sufficient for transport motors such as KIF5C to attain MT depolymerization function.

## Characterization of KIF19A ATPase stimulated by MTs and tubulins

As described above, KIF19A not only depolymerizes MTs mainly from their plus-ends, but can also move slowly along MTs (*Niwa et al., 2012*). Therefore, it is expected that KIF19A ATPase will be activated by both MTs and tubulins. To confirm this idea, we first examined the steady-state ATPase activity of the wild-type construct in the absence/presence of MTs or tubulins. The basal ATPase activities of 353WT, L55A, and PC2A in the absence of MTs or tubulins were all comparable to that of another kinesin-8 motor, Kip3p (*Figures 4A* and *Figure 4—figure supplement 1*) (*Gupta et al., 2006*). Activation of 353WT by MTs caused the ATPase to be activated ~300 times to reach a maximum rate of $1.43 \pm 0.07$ s$^{-1}$, while free tubulin-dimers caused the ATPase to be activated ~200 times to reach a maximum rate of $0.80 \pm 0.06$ s$^{-1}$ (*Figures 4A* and *Figure 4—figure supplement 1*). Thus, the KIF19A ATPase is similarly activated by both MTs and tubulins. This characteristic is conserved

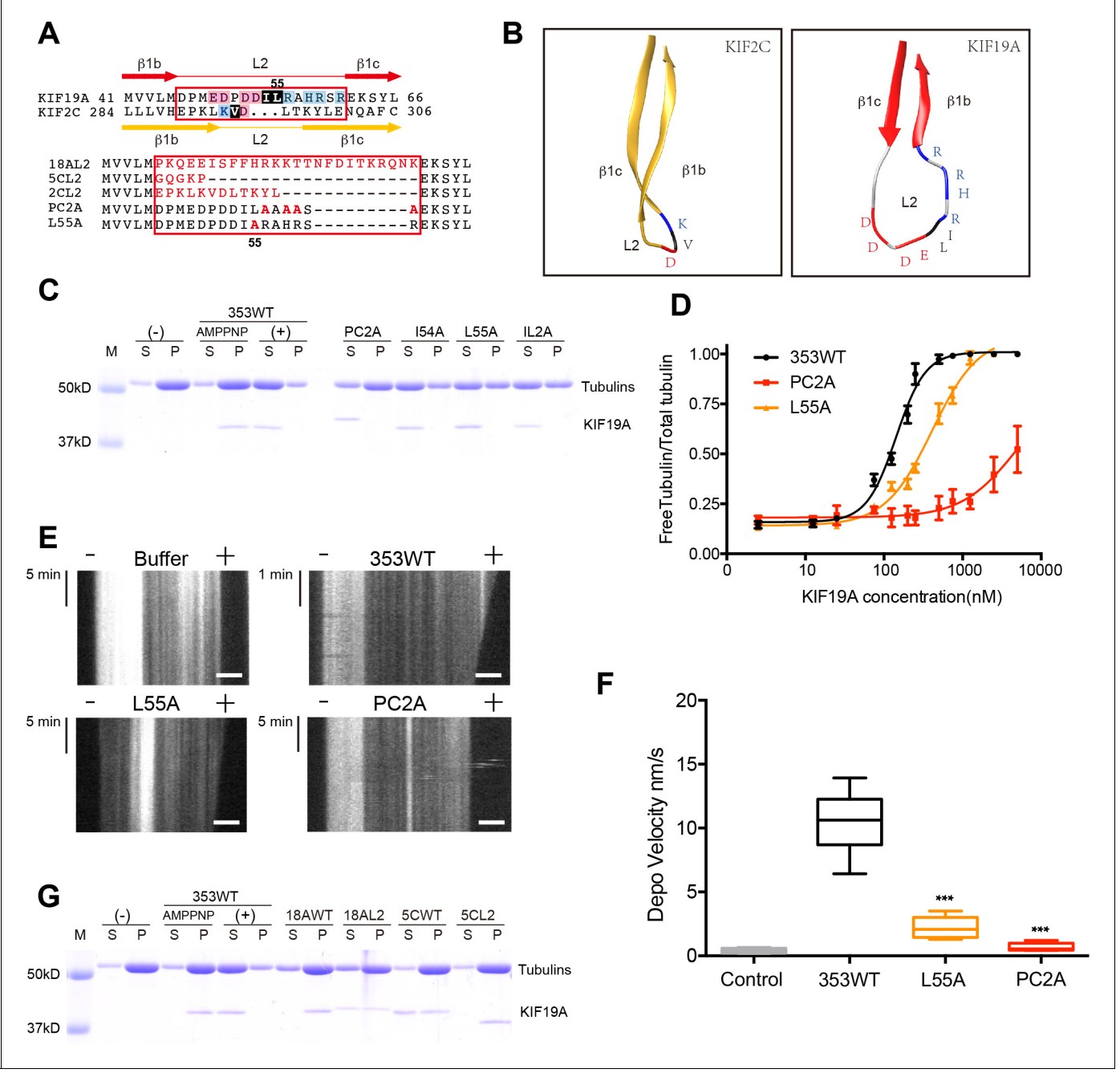

**Figure 3.** Basic and hydrophobic residues in L2 contribute to MT-depolymerizing activity. (A) Top, sequence alignment of L2 between KIF19A and KIF2C. Bottom, the L2 swap mutation and other point mutation strategies are shown. (B) β1b-L2-β1c structure diagram of KIF2C (gold) and KIF19A (red). The acidic, hydrophobic and basic residues are colored red, black and blue, respectively. (C) 1.5 μM GMPCPP-stabilized MTs were co-sedimented with 250 nM 353WT or its L2 mutants in the presence of 5 mM ATP or AMP-PNP. 5 mM AMP-PNP was used as a control. (D) Dose-response MT depolymerization curve for different concentrations of 353WT and its L2 mutants in the presence of 5 mM ATP. Data are presented as the mean ± SD. The mean EC$_{50}$ values of 353WT, PC2A and L55A were 142 ± 2 nM, 4936 ± 15 nM and 409 ± 4 nM, respectively. The data from three independent experiments were analysed. (E) Kymographs from MT depolymerization. Segment-marked microtubules incubated with 125 nM 353WT or its L2 mutants. Scale bars, 2 μM (horizontal) and time (vertical). (F) The depolymerization speeds of 353WT, L55A and PC2A were 10.9 ± 2.0 nm/s, 2.5 ± 0.5 nm/s, and 0.7 ± 0.2 nm/s, respectively. Each column represents the mean ± SEM (n = 45, 67, 58 and 51 MTs for buffer, 353WT, L55A and PC2A, respectively. Data were analysed using a two-tailed t-test, ***p<0.001, compared with 353WT). (G) 1.5 μM GMPCPP-stabilized MTs were co-sedimented with 250 nM 353WT, KIF18A, KIF5C, and the swap mutants, whose L2 loops were replaced by the counterpart of KIF19A. 5 mM AMP-PNP was used as a control. Representative data from three independent sample preparations are shown.

*Figure 3 continued on next page*

*Figure 3 continued*

The following source data is available for figure 3:

**Source data 1.** The data and analysis for 353WT and L2 mutants.

among kinesin-8 proteins, including KIF18A (*Peters et al., 2010*). However, the ratio of KIF19A ATPase activation by tubulins/MTs is much higher than that of KIF18A, suggesting that KIF19A is more adapted to tubulins to increase MT depolymerization activity.

Compared to KIF19A 353WT, L55A exhibited relatively small effects on ATPase turnover in the absence or presence of MTs (*Figures 4A* and *Figure 4—figure supplement 1C*), or in the presence of tubulins (*Figures 4A* and *Figure 4—figure supplement 1F*). The affinities for MTs and tubulins were slightly affected by the L55A mutation (*Figure 4A and 4B*). This indicates that the role of L55 in MT depolymerization is independent from that of the KIF19A catalytic cycle stimulated by MTs or tubulins. This L55 property seems similar to that of the valine residue of the KVD finger in kinesin-13 (*Ogawa et al., 2004*). In contrast, PC2A significantly decreased the MT/tubulin stimulation of the KIF19A ATPase (*Figure 4A* and *Figure 4—figure supplement 1*). The PC2A mutation did not significantly affect the $K_{M, tubulin}$, but markedly increased the $K_{M, MT}$ to approximately 30 times that of 353WT (*Figures 4A* and *Figure 4—figure supplement 1*), suggesting that the basic residues in L2 might strengthen the MT-lattice binding at the weak-binding state, similar to the K-loop of KIF1A (*Okada and Hirokawa, 2000*). The PC2A mutation also decreased the ATPase stimulation by tubulin without affecting the $K_{M, tubulin}$. Hence, the KIF19A-specific basic residues of L2 might be involved in the tubulin-activated ATPase pathway, as described later.

## Loop L2 tethers KIF19A to MTs and might slow down motility

The alanine mutation in the basic L2 cluster in KIF19A weakens the affinity for MTs. This was confirmed by a MT co-sedimentation assay using KIF19A and a non-hydrolysable ATP-analogue, adenylyl-imidodiphosphate (AMP-PNP). We found that L55A slightly decreased MT affinity (353WT, $K_{d, MT}$ 0.6 ± 0.1 μM; L55A, $K_{d, MT}$ 1.6 ± 0.3 μM), whereas PC2A severely decreased MT affinity (PC2A, $K_{d,MT}$ 5.8 ± 2.1 μM), consistent with the kinetics data above (*Figure 4B*). We also observed the MT gliding motility of KIF19A. The speed of the L55A mutant was a little higher than that of 353WT described above (L55A, 7.5 ± 0.6 nm/s, *Figure 4C and D*). However, bound MTs were rarely detected when the PC2A mutant, at the same concentration as L55A, was fixed on the coverslip. We thus checked the number of bound MTs to PC2A in the presence of AMP-PNP. Different concentrations of MTs were incubated with 353WT or PC2A in the presence of 5 mM AMP-PNP. Although PC2A bound fewer MTs than 353WT, increasing the concentration of PC2A (1 mg/ml) enabled sufficient microtubules to fix to the glass surface (*Figure 4—figure supplement 2*). Even in these conditions, however, bound MTs glided distances that were too short to reliably calculate the gliding speed on the PC2A-coated coverslip. Thus, the PC2A mutation severely increased the dissociation constant for MTs so that few MTs were observed on PC2A attached to the coverslip. The KIF18A and KIF5C L2 substitution mutations were also introduced into KIF19A to check MT gliding motilities. Again, bound MTs were not detectable with KIF5C L2 and very few MTs were detected with KIF18A L2, indicating that the L2 of KIF19A, especially its basic cluster, is necessary for 353WT to stay attached to MTs.

We further investigated the role of loop L2 for MT-based motility by the reverse swapping of KIF19A L2 into other kinesin motors. The L2 of KIF19A was swapped into the kinesin-8 member, KIF18A, and the kinesin-1 member, KIF5C, to clarify its effect on motility. As a consequence, MTs glided much more slowly with the swap mutants compared with the wild-type motors (KIF18A WT, 34.4 ± 0.3 nm/s; KIF18A swap, 10.2 ± 0.1 nm/s; KIF5C WT, 179.9 ± 2.5 nm/s; KIF5C swap, 68.9 ± 0.2 nm/s) (*Figure 4E–H*). The intensive ionic interactions between the L2 of KIF19A and MTs might reduce motility speed. KIF19A L2 is thus likely to interact electrostatically with the MT to tether the KIF19A motor domain on the MT, which in turn reduces motility speed on the MT.

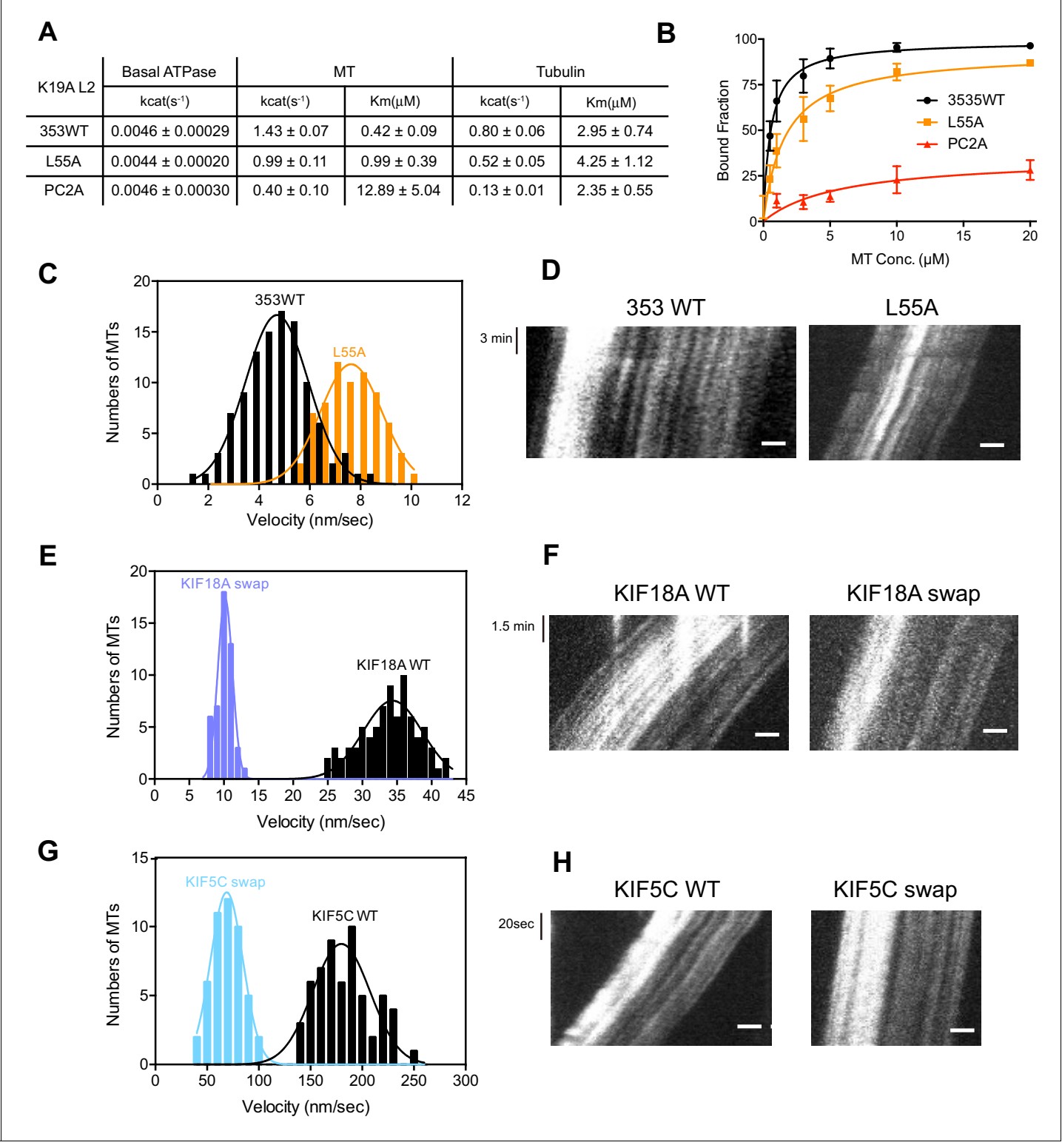

**Figure 4.** L2 also affects KIF19A kinetics and motility. (**A**) Steady state ATPase kinetics of 100 nM 353WT and its L2 mutants. Data are presented as the mean ± SD (n = 3). (**B**) Graph of the bound fraction of 353WT and L2 mutants plotted against the concentration of MTs. Fitting the data to a one site binding model gave a dissociation constant ($K_d$). Data are presented as the mean ± SD (n = 3). (**C**) MT sliding speed on 353WT and the L55A mutant. Data are presented as the mean ± SEM (n = 69 MTs for L55A). (**D, F, H**) Kymographs show MTs gliding on L55A and other constructs as imaged by TIRF microscopy. Scale bars, 2 μM (horizontal) and time (vertical). (**E, G**) Introduction of KIF19A L2 into KIF18A (**E**) and KIF5C (**G**) to generate two swap

*Figure 4 continued on next page*

*Figure 4 continued*

mutants. MT sliding speed of the wild type and swap mutants. Data are presented as the mean ± SEM (n = 80 and 58 MTs for KIF18A WT and KIF18A swap, respectively, n = 60 and 48 MTs for KIF5C WT and KIF5C swap, respectively).

The following source data and figure supplements are available for figure 4:

**Source data 1.** The data and analysis for 353WT, L2 and swap mutants.

**Figure supplement 1.** ATPase kinetics of 353WT and Its L2 mutants.

**Figure supplement 2.** Binding affinity difference between 353WT and PC2A.

## Basic residues in loop L12 increase the affinity for both tubulins and MTs to enable effective depolymerization of MTs

The most remarkable structural difference between KIF19A and other kinesins was found in the switch II region, α4-L12-α5, which is destabilized and distant from the motor domain (*Figures 2A–B* and *5B*). Considering that switch II serves as a major interface for MTs and tubulins, this conformational difference is expected to contribute to the dual functions of KIF19A. To address this question, we investigated the specific roles of the KIF19A switch II region in motility and depolymerization activity.

A sequence alignment of the switch II cluster in KIF19A, KIF5C and KIF18A shows marked differences that are concentrated in loop L12. Thus we first constructed two KIF19A swap mutants, in which L12 was replaced with the counterpart of KIF18A or KIF5C (*Figure 5A*). In comparison to 353WT, soluble tubulin was decreased in the presence of KIF5C L12, but was slightly increased in the presence of KIF18A L12 (*Figure 5C*). Within the KIF18A L12 sequence, three continuous basic residues (K299, R300 and R301) are present. Only one basic residue, K274, was found in the L12 of KIF5C, whereas two basic residues, K290 and K294, were found in the L12 of KIF19A. Therefore, to test the relationship between the number of basic residues in L12 and MT depolymerization ability, we made two more constructs: GS2R, in which both G291 and S292 were mutated to arginine, such that L12 had four basic residues, and K2A in which both K290 and K294 were replaced with alanine, such that L12 had no basic residues. As expected, GS2R exhibited a slight increase in MT depolymerization activity, similar to the KIF18A L2 swap mutant, whereas K2A without basic residues in L12 showed lower MT depolymerization ability, similar to the KIF5C L2 swap mutant (*Figure 5D–G*).

We also examined the ATPase of L12 mutants, GS2R and K2A. The basal ATPase activities of L12 mutants in the absence of MTs or tubulins were all comparable to the 353WT (*Figure 5—figure supplement 1*). The GS2R mutant showed only a small increase in MT- and tubulin-activated turnover rate, though it exhibited a four-times lower Michaelis-Menten constant for both MTs and tubulins (*Figure 5H*). In contrast, K2A showed a lower MT- and tubulin-activated turnover rate and a higher Michaelis-Menten constant for both MTs and tubulins (*Figure 5H*). Therefore, the change in the number of basic residues in L12 might alter the electrostatic affinities to the MT lattice or tubulins. To further confirm this idea, a MT co-sedimentation assay with Taxol-stabilized MTs was performed. The GS2R mutation did not significantly alter the affinity for MTs (353WT, $K_{d,\ MT}$ 0.6 ± 0.1 μM; GS2R, $K_{d,\ MT}$ 0.6 ± 0.06 μM). The K2A mutation, however, significantly weakened the affinity for MTs ($K_d$ 3.0 ± 0.6 μM) (*Figure 5I*). Thus, two basic residues of L12 contribute to the depolymerization ability of KIF19A through electrostatic interaction. Further addition of basic residues did not effectively increase MT affinity.

This contribution of basic residues in loop L12 resembles that of the KIF1A K-loop (*Okada and Hirokawa, 2000*). Thus, the binding partner of KIF19A L12 was expected to be the E-hook, the C-terminal tail of tubulin that has an acidic cluster. To address whether the E-hook is responsible for the electrostatic interaction with the KIF19A motor domain, we incubated KIF19A with two kinds of MTs, normal tubulin polymers and tubulin-S polymers, in the presence of AMP-PNP. Tubulin-S, which was obtained by limited subtilisin proteolysis of tubulin dimer, lacks the cluster of negatively charged residues found in the E-hook of α- and β-tubulin. Whether the binding ratio was 1:1 or 1:2.5, a

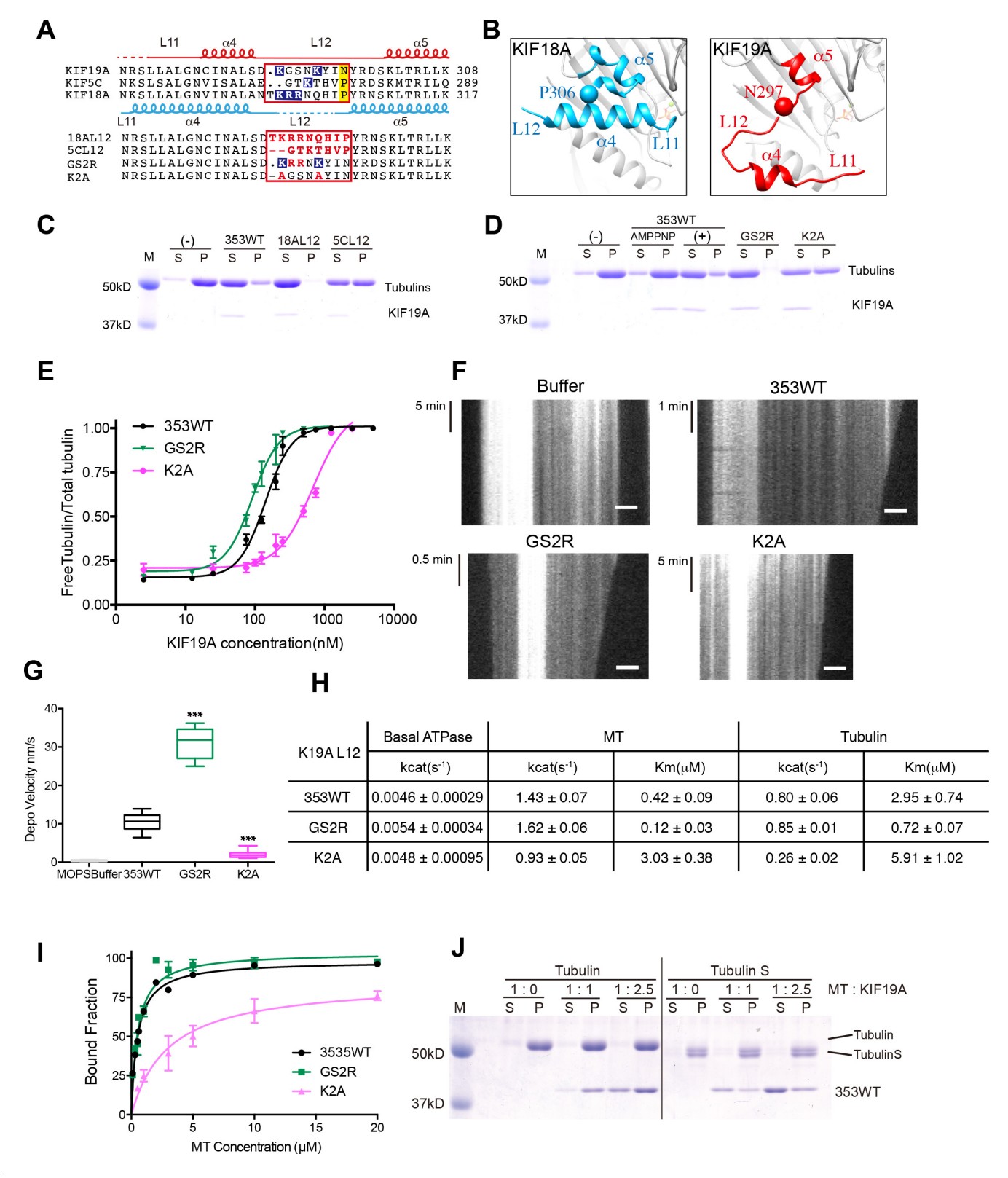

**Figure 5.** Basic residues in L12 assist KIF19A to effectively depolymerize microtubules. (**A**) Top, sequence alignment of KIF19A, KIF5C and KIF18A. Bottom, the L12 swap mutation and other point mutation strategies are shown. (**B**) L11-α4-L12-α5 structure of KIF18A (light blue) and KIF19A (red). KIF18A P306 and KIF19A N297 are shown as balls. (**C**, **D**) 1.5 µM GMPCPP-stabilized MTs were co-sedimented with 250 nM 353WT or its L12 swap

*Figure 5 continued on next page*

*Figure 5 continued*

mutants (C) and charged mutants (D) in the presence of 5 mM ATP or AMP-PNP. AMP-PNP (5 mM) was used as a control. (E) Dose-response MT depolymerization curve for different concentrations of 353WT and its L12 mutants in the presence of 5 mM ATP. Data are presented as the mean ± SD. The mean $EC_{50}$ values of 353WT, GS2R and K2A were 142 ± 2 nM, 91 ± 2 nM and 678 ± 5 nM, respectively. The data from three independent experiments were analysed. (F) Kymographs from MTs depolymerization. Segment-marked MTs incubated with 125 nM 353WT or its L12 mutants. Scale bars, 2 μM (horizontal) and time (vertical). (G) The depolymerization speeds of 353WT, GS2R and K2A were 10.9 ± 2.0 nm/s, 30.0 ± 4.9 nm/s, and 2.0 ± 0.5 nm/s, respectively. Each column represents the mean ± SEM (n = 45, 67, 52 and 47 MTs for buffer, 353WT, GS2R, and K2A, respectively). Data were analyzed using the two-tailed t-test, ***p<0.001, compared with 353WT). (H) Steady state ATPase kinetics of 100 nM 353WT and its L12 mutants. Data are presented as the mean ± SD (n = 3). (I) Graph of the bound fraction of 353WT and L12 mutants plotted against MT concentration. Data are presented as the mean ± SD (n = 3). (J) MTs polymerized by tubulin and tubulin S were co-sedimented with different concentrations of KIF19A in the presence of 1 mM AMP-PNP. Representative data from three independent sample preparations are shown.

The following source data and figure supplement are available for figure 5:

**Source data 1.** The data and analysis for 353WT and L12 mutants.
**Figure supplement 1.** Steady state ATPase kinetics of 353WT and Its L12 mutants.

greatly reduced quantity of KIF19A was seen in the tubulin S assembled MT pellet (*Figure 5J*). Thus, the E-hook of tubulin is confirmed to be responsible for the KIF19A-MT interaction.

## Asparagine 297 contributes to interface flexibility by destabilizing switch II helices to enable both MT and tubulin binding

We also investigated the asparagine in KIF19A at the junction of loop L12 and helix α5 (*Figure 6A*). At this position, a proline residue is highly conserved among kinesin superfamily proteins except for KIF19A (*Figure 6—figure supplement 1A*). It has been proposed that this proline serves as a 'helix starter' of α5 because of its restricted dihedral angles. Thus, a lack of proline at the starting position of α5 might destabilize helix α5, further destabilizing the preceding L12 and α4, as is observed in the crystal structure of 353WT in the absence of a MT (*Figure 2A and B*). To test this hypothesis, Asn297 was mutated to proline to clarify its effect on depolymerization and motility along MTs (*Figure 6C–G*). The resulting $EC_{50}$ value of N297P for MT depolymerization ($EC_{50}$ 201 ± 2 μM, *Figure 6C*) was slightly increased compared to the wild type, with the slight decrease in the depolymerization speed (7.7 ± 1.3 nm/s, *Figure 6D and E*). In contrast, the MT-gliding velocity (6.5 ± 0.2 nm/s) was slightly increased compared to the wild type in three independent experiments (*Figure 6F and G*). Therefore, Asn297 in α5 has a slight advantage for the MT depolymerization but has a slight disadvantage for the motility over Pro297.

We then checked the basal ATPase activity as well as stimulation by MTs or tubulins. Even though no significant difference was shown, the N297P mutation tended to increase the basal ATPase activity (*Figures 6H* and *Figure 6—figure supplement 2*). The destabilized conformation of switch II might be less favorable for the basal ATPase activity. The MT-stimulated ATPase rate did not alter significantly, although the Michaelis-Menten constant was increased (*Figures 6H* and *Figure 6—figure supplement 2*). In contrast, the tubulin-stimulated ATPase rate of N297P was markedly reduced to approximately half that of the wild type (*Figures 6H* and *Figure 6—figure supplement 2*). Hence, the KIF19A-specific asparagine residue, Asn297, contributes to the increase of tubulin-stimulated ATPase activity but has minimal effect on MT-stimulated ATPase activity. Thus, Asn297 of KIF19A might be advantageous to adopting both the straight tubulin-dimer in the MT-lattice and the curved tubulin-dimer at the MT-ends. Asn297-induced destabilization of switch II helices might contribute to this adaptation to achieve the dual functions of KIF19A.

## Cryo-EM reconstruction of 353WT complexed with GDP-taxol MTs reveals the functional anatomy of KIF19A

Finally, we performed cryo-EM analysis of KIF19A in the nucleotide-free state complexed with a GDP-taxol MT. Cryo-EM reconstruction at a 7.0 Å resolution clarified the interactions between KIF19A specific structures and a straight MT (*Figure 7—figure supplement 1A*). It should be noted that, presumably because of the weak or unstable interaction between KIF19A and a straight MT,

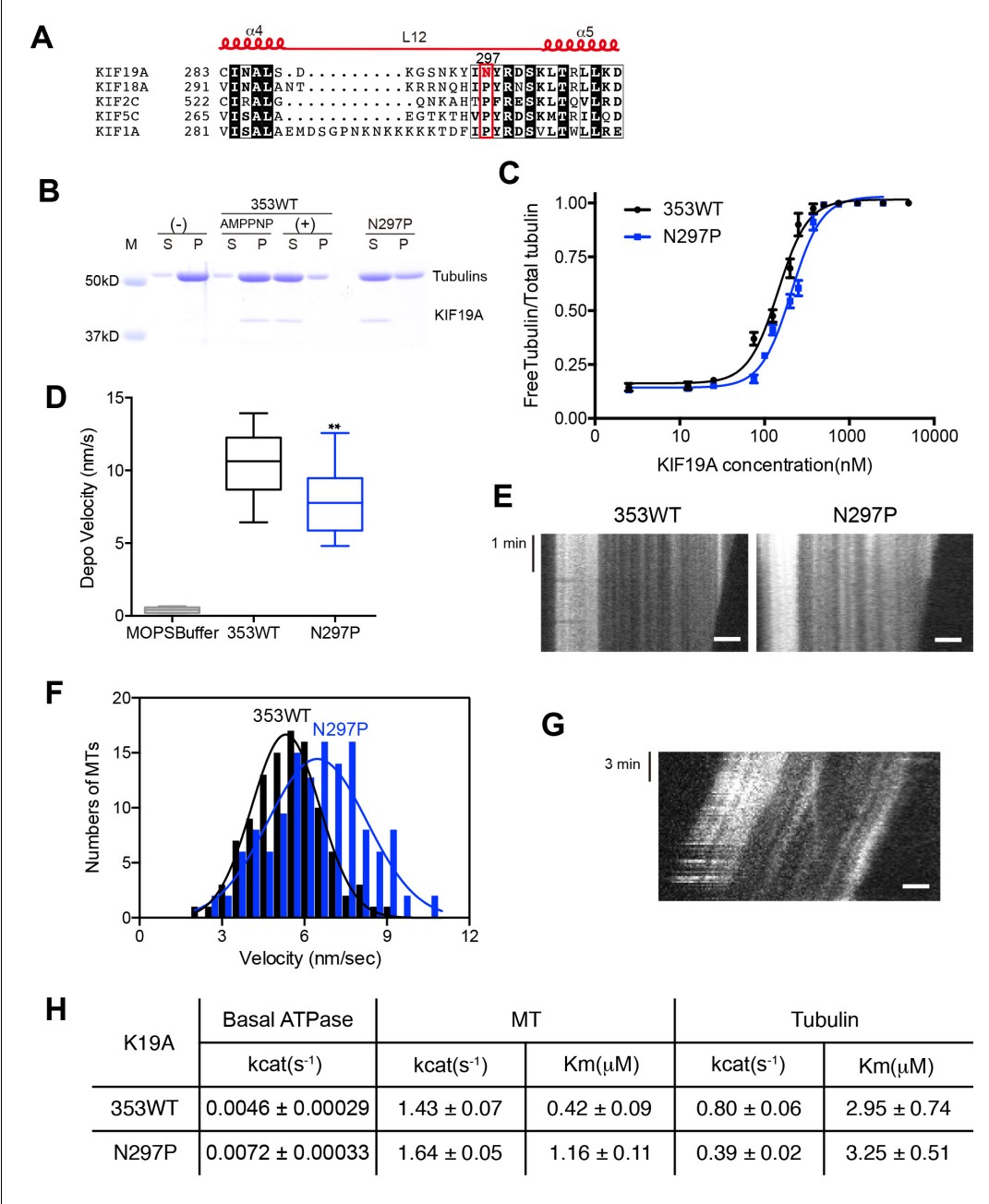

**Figure 6.** Contribution of KIF19A-Specific Asn297. (**A**) Sequence alignment of KIF19A and other typical kinesins for the α4-L12-α5 region. For more kinesin member sequence alignments see *Figure 6—figure supplement 1A*. (**B**) 1.5 μM GMPCPP-stabilized MTs were co-sedimented with 250 nM 353WT or N297P mutant in the presence of 5 mM ATP or AMP-PNP. Representative data from three independent sample preparations are shown. AMP-PNP was used as a control. (**C**) Dose-response MT depolymerization curve for different concentrations of 353WT and N297P in the presence of 5 mM ATP. Data are presented as the mean ± SD. The mean $EC_{50}$ values of 353WT, N297 were 142 ± 2 nM and 201 ± 2 nM, respectively. The data from three independent experiments were analyzed. (**D**) The depolymerization speeds of 353WT and N297 were 10.9 ± 2.0 nm/s and 7.7 ± 1.3 nm/s, respectively. Each column represents the mean ± SEM (n = 67 and 29 MTs for 353WT and N297P, respectively). Data were analyzed using the two-tailed t-test, **p<0.01, compared with 353WT). (**E**) Kymographs of MT depolymerization. Segment-marked MTs incubated with 125 nM 353WT or N297P mutant. Scale bars, 2 μM (horizontal) and 1 min (vertical). (**F**) MT gliding assays of KIF19A-353WT and N297P on taxol-stabilized MTs. Data are presented as the mean ± SEM, (n = 113 MTs for N297P). (**G**) Kymograph showing movement of N297P long MTs as imaged by TIRF microscopy. Scale bars, 2 μM (horizontal) and 3 min (vertical). (**H**) Steady state ATPase kinetics of 100 nM 353WT and N297P. Data are presented as the mean ± SD, (n = 3).

The following source data and figure supplements are available for figure 6:

*Figure 6 continued*

**Source data 1.** The data and analysis for 353WT and N297P mutant.
**Figure supplement 1.** Sequence alignment between representative kinesin members.
**Figure supplement 2.** Steady state ATPase kinetics of 353WT and N297P mutant.

stable and full decoration of KIF19A on the GDP-taxol MT was uncommon; the occupancy of KIF19A is thus lower than that of a tubulin-dimer in our reconstruction (*Figure 7—figure supplement 1B and C*). However, almost all helices, β-sheets and important loops that contact a MT are reliably visualized in our reconstruction (*Figure 7—figure supplement 1D*).

To clarify the conformation of the nucleotide-free KIF19A on a MT, two crystal structures, the KIF19A-ADP structure solved here (*Figure 7A and B*) and the nucleotide-free KIF5 structure (PDB: 4LNU) (*Figure 7C and D*) (*Cao et al., 2014*), were rigidly fitted into our reconstruction. The KIF5-free structure represented better fitting than the KIF19A-ADP structure, as indicated by the higher cross correlation values (KIF19A-ADP, 0.83; KIF5-free, 0.89). Most of the regions in the catalytic core of both KIF19A-ADP and KIF5-free are nicely fitted into the cryo-EM map of KIF19A-free on a MT (gray in *Figure 7A–D*). However, marked differences were observed between the two structures at the MT-binding interfaces, loop L8 at the plus-end side, switch II α4-L12-α5, and loop L2 at the minus-end side (green or red in *Figure 7A–D*). The first two of these regions of the cryo-EM map take very similar conformations to those of KIF5-free on a MT (red arrows in *Figure 7C*), whereas the L2 of KIF19A-ADP was nicely fitted into the extra-density observed at the minus-end side (black arrow in *Figure 7A*). Thus, we finally created an atomic model of KIF19A-free on a MT that has the similar L8 and switch II conformations as KIF5-free and the similar L2 conformation as KIF19A-ADP (*Figure 7E*). This model exhibits the highest cross correlation value of 0.90 with our cryo-EM structure.

The L8 of KIF19A-free on a MT moves to interact with H12 residues of β-tubulin during binding with the MT-lattice and the release of ADP from the catalytic core (*Figure 7G*). This movement is accompanied by approximately 7 degree counter-clockwise rotation of the following helix α3, as also reported in kinesin-13 (*Figure 7H*) (*Ogawa et al., 2004*). Thus, the interaction with the straight lattice of the MT and the MT-induced ADP release from KIF19A triggers the counter-clockwise rotation of the L8-α3 complex (*Figure 7H*). The KIF19A conformation on the MT-lattice is similar to typical plus-end directed kinesins, such as KIF5, and might close the nucleotide-binding pocket even on a straight MT, allowing access for ATP into the pocket.

The switch II α4-L12-α5 of KIF19A-free is retracted to take a similar conformation to that of KIF5-free during MT-lattice binding and ADP release (*Figure 7G*). The length, the location, and the angle of α4 in KIF5-free fit perfectly to the corresponding cryo-EM density of KIF19A-free (*Figure 7F*). This means that KIF19A is able to adjust the switch II conformation to fit the straight tubulin interface on the MT-lattice. It is also notable that a clear density was found at the right side of α4, corresponding to the E-hook of β-tubulin (*Figure 7E and F*). As expected by the biochemical assays described above, the acidic residues in the E-hook ionically interact with the basic cluster of L12 in KIF19A, thus stabilizing the E-hook. In comparison with the E-hook of α-tubulin, that of β-tubulin is remarkably stabilized by interaction with L12 of KIF19A (*Figure 7E*).

Loop L2 extends toward the minus-end of the MT. No marked difference was observed between KIF19A-ADP and KIF19A-free on a MT (*Figure 7G*). The corresponding density of L2 almost reaches the inter-tubulin-dimer groove close to the H11'-helix of β-tubulin at the minus-end side (*Figure 7I*). Because L2 has the characteristic sequence in which the hydrophobic residues at the tip of L2 are sandwiched by an acidic cluster on the left side and a basic cluster on the right side, the possible interactions of these residues were estimated from the fitted atomic models of KIF19A and tubulin-dimers (*Figure 7J*). The acidic cluster faces the H11' helix of the neighboring β-tubulin, where basic H406 and K402 residues are located (red region in *Figure 7J*). However, the Cα distance between L2 and H11' is around 11 Å, which is slightly longer than a stable interacting distance. Hydrophobic residues presumably interact with the hydrophobic residues of the H8-S7 loop and the H12 helix in α-tubulin (yellow region in *Figure 7J*). Consistent with the mutational studies, the L55 of L2 serves as

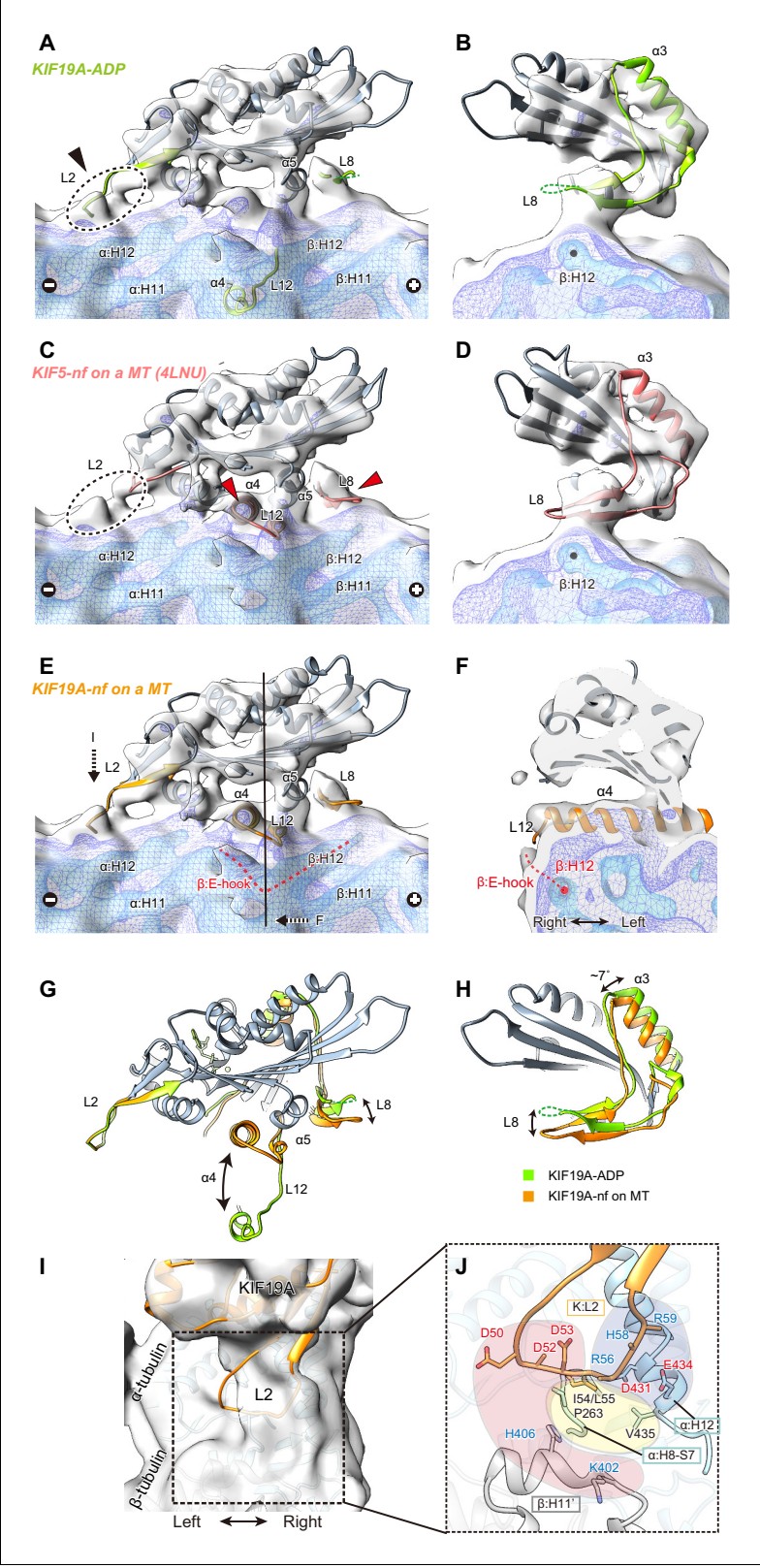

**Figure 7.** Cryo-EM reconstruction of KIF19A on a straight MT. (**A–D**) Cryo-EM reconstruction of KIF19A-nucleotide-free complexed with GDP-taxol-MT with three different contour levels (grey, blue, cyan), and atomic models of KIF19A-ADP solved in this study (green) (**A** and **B**) and KIF5-nucleotide-free (red, PDB: 4LNU) (**C** and **D**), seen from the right side (**A** and **C**) and the plus-end side (**B** and **D**). Black arrow and dashed circles indicate the

*Figure 7 continued on next page*

*Figure 7 continued*

elongated L2. Red arrows show the α4-L12 and L8 fitted in the corresponding densities. (E) Atomic model of KIF19A-nucleotide-free on GDP-taxol-MT (orange) was created and rigidly fitted onto the cryo-EM map. The density of the helix H12 and the following E-hook of β-tubulin can be observed (red dashed line), while the E-hook of α-tubulin is not clear. (F) Conformation of the switch II (α4–L12) and the E-hook of β-tubulin (red dashed line). Atomic model of α4 fits perfectly in the density. (G and H) Comparison of the KIF19A-ADP (green) and KIF19A-nucleotide-free models on MT (orange), viewed as in A and B. (I) L2 density was clearly observed at the inter-tubulin dimer interface. (J) Zoom-in view around L2. The acidic residues in L2 potentially interact with the basic residues in β-tubulin H11′ (red region). The basic residues in L2 interact with the acidic residues in α-tubulin H12 (blue region). Between these electrostatic clusters, the hydrophobic residues of L2 in KIF19A and H12 and H8-S7 loops in α-tubulin constitute the hydrophobic core (yellow region).

The following figure supplement is available for figure 7:

**Figure supplement 1.** Cryo-EM reconstruction of KIF19A-MT complex.

the main interface of the hydrophobic contacts. The basic cluster then faces acidic clusters on the H12 helix of α-tubulin (blue region in *Figure 7J*). Its Cα distances are within 9 Å, thus salt-bridges might be formed between them even on the straight MT lattice. When KIF19A is on the MT-lattice, therefore, loop L2 interacts with α-tubulin through the hydrophobic tip and basic cluster, though the acidic cluster will not contribute to the interaction with the MT.

## Discussion

As a unique subfamily of the kinesin superfamily, kinesin-8 members possess both MT-based plus-end directed motility and MT depolymerizing activity (*Gupta et al., 2006*; *Varga et al., 2006*). KIF18A and kip3p are the most extensively studied kinesin-8 members; however, KIF19A may provide a simpler model because a single motor domain of KIF19A intrinsically possesses the dual activities of MT-based motility and MT-depolymerizing activity. In this study, we clarified the functional anatomy of the KIF19A motor domain and propose a structural model for its dual functions.

### Two requirements for the dual functions of KIF19A

To achieve the dual functions, KIF19A should meet the following two requirements: (1) the ability to adopt two different interfaces for straight MTs and curved tubulins to enable KIF19A ATPase activation, and (2) the ability to stabilize the curved conformation of MT-ends to destabilize the MT proto-filaments, as observed for kinesin-13 (*Figure 8A*) (*Ogawa et al., 2004*; *Shipley et al., 2004*). To accomplish these requirements, KIF19A-specific characteristics are concentrated on the MT-binding surface. As detailed in the following sections, for the 1st requirement, the location of the destabilized, retractable loop L8 and switch II (α4-L12-α5) enable adjustment to both the straight tubulin interface on the MT-lattice [*Figure 8C(i)*] and the curved tubulin interface at the MT-ends [*Figure 8C (iv)*]. For the second requirement, the long and fan-shaped L2 is located at the minus-end side. It plays a central role in depolymerizing MTs from the ends through stabilization of the curved conformation of MT-ends. Helix α6 is atypically shorter than those of the other previously solved kinesins. Its functional meaning is still uncertain mainly because structural information for the neck-linker that follows α6 is still lacking. The short conformation of α6 is expected to affect the neck-linker conformation and further structural study of KIF19A in the ATP state is needed to address the role of the short helix α6.

### L8 and switch II of KIF19A bring flexibility to fit the tubulin interfaces in both the MT-lattice and MT-curved end

Tubulin-dimers have distinct conformations when located in the MT-lattice or in the MT-curved end (*Figure 8B*). Most kinesins choose between the two, although kinesin-8 members are able to bind both. To investigate which structural alterations are needed to achieve this, in silico docking of KIF19A with a curved tubulin-dimer (PDB: 3RYC) was performed (*Nawrotek et al., 2011*). The α-tubulin of the curved tubulin-dimer was aligned with that of the straight MT-lattice.

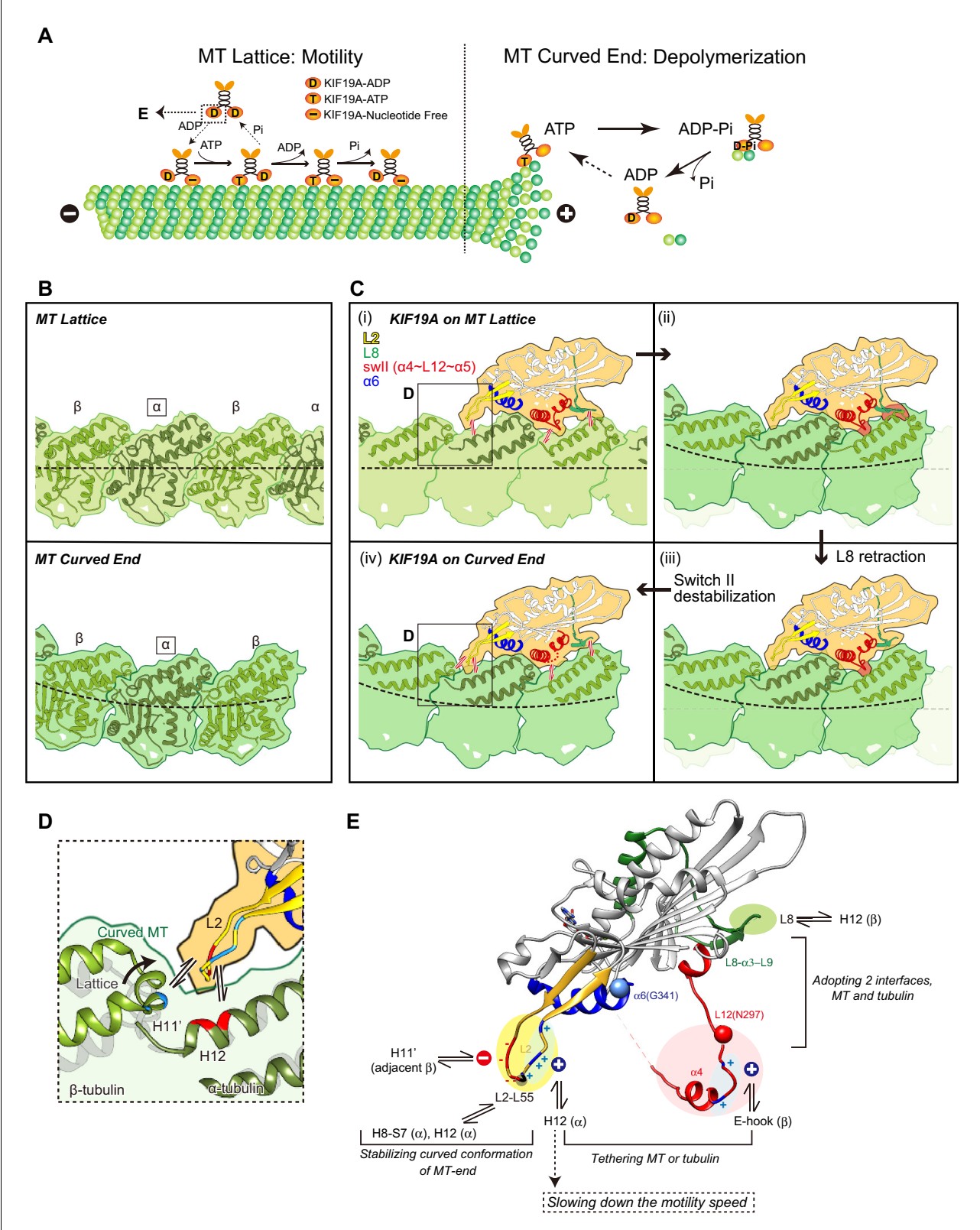

**Figure 8.** Model of KIF19A motor domain function. (**A**) Schematic diagram of the dual functions of KIF19A. (**B**) Simulated 7 Å maps of straight and curved MTs created from the atomic models of 1JFF and 3RYC, respectively. The two structures were aligned at the position of the marked α-tubulins. (**C**) Simulated 7 Å maps and atomic models of KIF19A on straight and curved MTs showing the adaptation of KIF19A-MT interfaces. Collisions between KIF19A and β-tubulin in (ii) and (iii) are highlighted in red. (**D**) Zoom-in view around the KIF19A L2 and curved tubulin-dimer interface (green). Atomic

*Figure 8 continued on next page*

Figure 8 continued
model of a straight MT (grey) is superimposed to compare the distance from L2 to β-tubulin H11'. (E) Functional anatomy for the dual function of KIF19A.

On the MT-lattice, the interface of the KIF19A-free model is nicely fitted to the tubulin-dimer interface [*Figure 8C(i)*]. L8, switch II, and L2 serve as main interfaces for the straight tubulin-dimer. The KIF19A-free model on the MT-curved end comes into collision with the β-tubulin surface at two regions, L8 and switch II L12-α5 [*Figure 8C(ii)*]. L8 in the KIF19A-ADP model, however, made better contact with the curved tubulin-dimer at the MT ends [*Figure 8C(iii)*]. Hence, the retractable L8 by the rotational movement of the L8-α3-L9 cluster is the first key feature to enable fitting to both the straight and curved interfaces of MTs.

However, switch II still collides with curved β-tubulin at the junction of L12 and the first turn of helix α5 [*Figure 8C(iii)*]. At that point, the highly conserved proline residue is replaced in KIF19A with a unique asparagine residue (Asn297). A proline residue is often found at the start or end points of a helix to stabilize it. Thus, replacement of proline for an asparagine residue might destabilize α5 allowing it to melt into the flexible long L12 loop. In fact, the first turn of helix α5 melted in the crystal structure (*Figure 2F*). Therefore, we expected that the flexible feature of the L12 and initial α5 might be caused by the KIF19A-specific Asn297. This idea was supported by our biochemical experiment. The N297P mutant had a more than 40% decrease in tubulin-stimulated ATPase activity compared with the wild type, indicating that this mutation limits adjustment of the KIF19A interface to the curved tubulin-dimer (*Figure 6H*). Thus, the destabilized conformation of switch II is the second key feature to avoid collision when binding to the curved MT end [*Figure 8C(iv)*]. To confirm this idea, a high resolution structure of KIF19A complexed with tubulin is required.

## KIF19A compensates for decreased MT affinity by flexible ionic interactions through L2 and L12

The arrangements of the KIF19A MT-binding interface described above enable it to achieve dual functions. On the flip side, however, they result in decreased affinity of KIF19A for the MT-lattice or the MT-curved end. KIF19A has overcome this difficulty by introducing basic clusters into the L2 and L12 loops.

The basic cluster at the right side of L2 tethers MTs by ionic interactions between L2 and the MT. The alanine mutant, PC2A, severely decreases the MT-binding affinity of KIF19A so that MTs were rarely found on the PC2A-bound coverslip. It should be noted here that the PC2A mutation only decreases the affinity for MTs but not for soluble tubulin-dimers (*Figure 4A*). As described below, the additional salt bridges through the acidic cluster on the left side of L2 are only formed when interacting with a tubulin-dimer. This could possibly minimize the effect of the PC2A mutation on the affinity for tubulins, but not on the affinity for MTs. In contrast, when the basic cluster was introduced into the L2 of KIF5C or KIF18A, the motility speed decreased to nearly one third of wild type. The additional strong interaction between the basic cluster and MTs might slow down the detachment of KIF19A from the MT. Hence introducing the basic cluster of L2 to increase affinity for MTs has the consequence of reducing the speed of KIF19A motility.

Similar to the basic cluster of L2, the basic residues of L12 contribute to tethering KIF19A to the E-hook of a MT, in a similar fashion to the K-loop of KIF1A, (*Okada and Hirokawa, 2000*). Our data showed that more than two of the basic residues in L12 were required to show wild-type KIF19A MT-binding ability and MT-depolymerization activity. In kinesin-13, alanine substitutions of charged residues in the switch II cluster showed lower depolymerization activity due to weak MT binding ability (*Shipley et al., 2004*). We presume that the positively charged residues support both straight and curved MT binding, providing minor assistance to L2.

## The structural mechanism of KIF19A MT depolymerization

The shapes of the fan-like L2 of KIF19A and the slender L2 of kinesin-13 were different, while the length of loop L2 is comparable between KIF19A and kinesin-13 and the locations of the hydrophobic residues at the tip are conserved (*Figure 3A*). Alanine mutation of Leu 55, which caused a MT

depolymerization deficiency with a small effect on MT-based motility, suggested its central role for depolymerizing MTs (*Figure 3*). The comparison of KIF19A binding to the MT-lattice and the MT-curved end indicates a further role for L2 [*Figure 8C(i) and (iv)*]. The interactions between L2 of KIF19A and α-tubulin through the hydrophobic tip of L2 and the basic cluster of L2 are similarly observed for both straight and curved tubulins. For curved tubulins, however, interactions between L2 and the neighboring β-tubulin at the minus-end side would also be formed (*Figure 8D*). The H11′ helix of β-tubulin gets closer to L2 by ~2Å compared with that in the straight conformation (Cα distance is around 9 Å) so that the acidic cluster could make salt bridges with the basic residues in the H11′ helix of β-tubulin. Therefore, the combination of acidic-hydrophobic-basic residues of L2 stabilizes the curved conformation of inter-tubulin-dimer interface. This could be the structural basis for MT depolymerization by KIF19A.

## Model of KIF19A motor domain function

Based on the structural properties of the KIF19A motor domain and substantial mutant analysis, we present a functional anatomy for the dual function of KIF19A (*Figure 8E*). The destabilized switch II helices contribute to the adaptation to the two distinct interfaces of straight and curved MTs with a concomitant sacrifice of slightly decreased motility speed along the MT. The rotatable L8-α3-L9 cluster also supports adaptation to the two interfaces. The basic cluster of L2, as well as the basic residues of L12, enable KIF19A to tether to both straight and curved MTs via flexible ionic interactions with the acidic residues of H12 or the E-hook of tubulins. This assures its motility along the MT, although motility speed may be decreased. The hydrophobic tip of L2, as well as the surrounding basic and acidic clusters, play critical roles in MT depolymerization. These residues stabilize the inter-tubulin-dimer interface of the curved MT protofilament. Thus, the curved conformation of MT ends is stabilized by L2, resulting in the depolymerization of the MTs. In this way, KIF19A has acquired dual functions by introducing multiple strategies. The resulting KIF19A is a slow plus-end directed motor combined with mild MT-depolymerizing activity. The KIF19A motility speed and the MT depolymerization activity might be different on ciliary doublet MTs. Further studies are required to reveal how KIF19A is involved in length control of ciliary MTs *in vivo*.

# Materials and methods

## Constructs and protein preparation

The coding sequence of the mouse KIF19A motor domain (1–353) and its related mutants were cloned into pET21b(+) (Novagen, Germany) with a 7×His-tag at the C terminal of the motor domain. All constructs were transformed into *E. coli* strain BL21(DE3) (Novagen). Protein expression was induced by the addition of 0.4 mM IPTG to cultures followed by incubation at 24°C for 16 hr with vigorously shaking. Immobilized-metal affinity chromatography (His-select Nickel Affinity Gel, Sigma-Aldrich, Saint Louis, MO) and ion exchange chromatography (Resource S, GE Healthcare, Japan) were sequentially used for protein purification. The protein samples were aliquoted in small volumes for immediate use or flash frozen in liquid nitrogen for later use.

## Crystallographic methods

The hanging drop vapor diffusion method was used. One microliter of KIF19A353WT protein sample at 10 mg/ml containing 0.1 mM ADP was mixed with 1 μl reservoir buffer and incubated at 20°C. Single Crystals are grown in 10% ethylene glycol, 2% PEG8000, 50 mM Tris-Bicine (pH 8.5), 9 mM $MgCl_2$ and 9 mM $CaCl_2$. X-ray diffraction data at 2.72 Å resolution were collected using a BL41XU beam-line (SPring-8, Japan), at a wavelength λ = 1.0 Å. The anomalous diffraction data were collected using a BL1A beamline (Photon Factory, Japan), at the wavelength λ = 2.7 Å. The HKL2000 program package (*Otwinowski and Minor, 1997*) was used to index, integrate and scale the data. The structure of 353WT was solved by molecular replacement using the Crystallography & NMR System (CNS) (*Brunger, 2007*; *Brünger et al., 1998*) and the atomic coordinates of KIF18AMD (PDB: 3LRE) as a search model. Several rounds of iterative model building and refinement were performed using COOT (*Emsley and Cowtan, 2004*) (RRID: SCR_014222) and Refmac5 (*Murshudov et al., 2011*) (RRID: SCR_014225). The final crystallographic model of 353WT at 2.7 Å resolution has a $R_{work}/R_{free}$ of 0.222/0.302. The data collection and refinement statistics are shown in

*Supplementary file 1*. UCSF Chimera (RRID: SCR_004097) was used for structure alignment and visualization (*Pettersen et al., 2004*).

## Microtubule gliding assay

Porcine brain tubulin was purified and tetramethylrhodamine (TMR) labeled. The microtubule gliding assay was performed as previously described with some modifications (*Niwa et al., 2012*). KIF19A 353WT and its mutants were immobilized on a coverslip using PentaHis antibody (Qiagen, Netherlands, RRID: AB_2619735). The surface of the coverslip was further coated with casein (Wako Chemical, Japan) to prevent nonspecific binding. After washing, 20 µl KIF19A protein solution (0.1 mg/ml) was injected into the flow chamber and washed out after 3 min incubation. Then, TMR-labeled MTs in PEM buffer (100 mM PIPES-KOH pH 6.8, 1 mM EGTA, 1 mM MgCl$_2$, 10 µM taxol) were injected and allowed to bind for 5 min. Taxol-stabilized MTs were used to minimize the depolymerization of MTs by KIF19A. The coverslip was then turned face down for 3 min to reduce background fluorescence and to prevent extra MTs binding during data collection. Finally, the motility buffer was supplemented with oxygen-scavenger just before observation to minimize photo bleaching. Time-lapse observation was performed at 37°C, using the ELYRA P.1 system (Carl Zeiss, Germany) in the TIRF mode. The data were collected every 10 seconds and tracking time was 15 min for 353WT and its mutants. The data were collected every 5 seconds and the tracking time was 7.5 min for KIF18A WT and KIF18A swap. The data were collected every 1 second tracking time was 7.5 min for KIF5C WT and KIF5C swap.

## Microtubule depolymerization assay

GMCCPP-stabilized MTs were polymerized as previously described (*Yajima et al., 2012*). Depolymerization assays were performed using GMPCPP MTs (*Hertzer et al., 2006*; *Niwa et al., 2012*; *Noda et al., 2012*). 250 nM wild-type or mutant KIF19A was incubated in BRB80 buffer with 5 mM ATP or AMPPNP and 1.5 µM GMPCPP MTs. To determine the EC$_{50}$, 353WT was titrated at different concentrations (0–5000 nM) with 1.0 µM GMPCPP MTs. All reactions were incubated at 25°C for 15 min, and subsequently centrifuged in a TLA55A rotor (Beckman Coulter, Brea, CA) at 38,000 ×g for 10 min at 25°C. The supernatant was removed from the pellet, and the pellet was resuspended in the same volume of BRB80 buffer as the supernatant. Equal volumes of supernatant and pellet were electrophoresed on a 10% SDS-PAGE. The gel was stained with Coomassie Brilliant Blue and scanned. The fraction of free tubulin was quantified and analyzed using ImageJ (NIH). The data were plotted and fitted to the four-parameter logistic equation and the EC$_{50}$ was calculated using Kaleida-Graph 4.0 software (Synergy software). Time-lapse observation of microtubule depolymerization was performed as previously described (*Niwa et al., 2012*).

## ATPase assay

The steady state ATPase kinetics of KIF19A 353WT and mutants (100 nM) were measured using an EnzCheck phosphate assay kit (Molecular Probes, Eugene,OR) (*Nitta et al., 2008*). In the presence of inorganic phosphate, P$_i$, whose production is catalyzed by Kinesin, the substrate 2-amino-6-mer-capto-7-methylpurine riboside (MESG) is enzymatically converted by purine nucleoside phosphory-lase (PNP) to ribose 1-phosphate and 2-amino-6-mercapto-7-methylpurine. This process results in a spectrophotometric shift in maximum absorbance to 360 nm for the product. The absorbance at 360 nm was recorded at 25°C every 5 s for 300 s using a V-630 Bio spectrophotometer (JASCO, Japan). The ATPase hydrolysis rate was calculated from the slope of the absorbance plot, and measured at different concentrations of MT or tubulin. Kinetic data were plotted and fitted the Michaelis-Menten model.

## Microtubule binding assay

Two micromoles of purified KIF19A protein were incubated with increasing amounts of MTs (0–40 µM) in PEM buffer at 25°C for 15 min. To minimize the MT depolymerization by KIF19A, taxol-sta-bilized MTs were used. Samples were centrifuged at 38,000 ×g for 10 min at 25°C, in a TLA55 rotor (Beckman Coulter). The separated supernatant and pellet fractions were loaded onto SDS-PAGE gels and resulting band intensities were analyzed using Image J (NIH). The data were fitted using

nonlinear regression and a one site-specific binding model. Dissociation constants ($K_d$) were calculated using KaleidaGraph 4.0 software (Synergy software).

## Tubulin S preparation

Tubulin was diluted to 5 mg/ml in distilled water and GTP added to 1 mM final concentration. After pre-incubating in a centrifuge tube at 25°C for 5 min, subtilisin was added in a weight ratio of 1/100, with further incubation at 25°C for 45 min. The reaction was stopped with PMSF to a final concentration of 0.05% and incubated on ice for 40 min. The samples were then centrifuged at 110,000 ×g at 4°C for 20 min (*Knipling et al., 1999*). The supernatant was collected and the protein concentration determined by the BCA method(ThermoFisher, Waltham, MA). Samples were used fresh or flash frozen in liquid nitrogen and stored at −80°C.

## Grid preparation and cryo-EM data collection

Tubulin (20 µM) was polymerized in polymerization buffer (PEM-GTP and 7% DMSO) at 37°C for 30 min. Taxol was added in a stepwise fashion to a final concentration of 20 µM. KIF19A 353WT was diluted to 100 µM in a dilution buffer (10 mM Hepes-NaOH pH 7.5, 50 mM NaCl, 1 mM MgCl$_2$). A 5 µl drop of the polymerized microtubules (4 µM) was placed onto a glow-discharging holey carbon grid (R2/2, Quantifoil). After 30 s, the solution was wicked away with a piece of Whatman no. 1 filter paper and a 5 µl drop of 353WT (28 µM) was quickly applied. After another 60 s, the grid was reacted with PEM containing 10 U/ml apyrase for 90 s and plunge-frozen into liquid ethane using a semi-automated vitrification device (Vitrobot Mark IV, FEI, Hillsboro, OR) with 5 s in 100% humidity at 27°C. Data acquisition was performed using a 200 kV field emission cryo-electron microscope (Tecnai Arctica, FEI) at 78,000-fold nominal magnification with an FEI Falcon II direct detection camera under low-dose conditions using the data acquisition software, Serial EM (*Mastronarde, 2005*). All data were collected as a movie with seven subframes with a total electron dose of 50 e$^{-1}$/Å$^2$ at a pixel size of 1.28 Å/pixel. The defocus range of the data set was set to −1.5 to −2.5 µm.

## Image processing and three-dimensional reconstruction of cryo-EM images

All the movie data were processed for motion correction using the software Unblur (*Grant and Grigorieff, 2015*). Motion corrected and summed images were analyzed for defocus and astigmatism using CTFFIND3 (*Mindell and Grigorieff, 2003*) and images without significant drift and astigmatism were used for further analysis. To generate an initial low-resolution model (~20 Å), images of a 14-protofilament motor-microtubule complex were selected and semi-automatically straightened using the 'unbend' program of Ruby-Helix (*Metlagel et al., 2007*). Three-dimensional structures were generated using asymmetric helical reconstruction (*Kikkawa, 2004*). This initial 3D structure was then used for single particle analysis as described previously (*Shang et al., 2014*). Segments were extracted at a spacing of 80 Å using a box size of 768 × 768 pixels and initial XY shifts and Euler angles were determined by template matching to the 3D model. The position of the microtubule seam was determined using the set of segment images extracted from one microtubule as described (*Sindelar and Downing, 2007*). Subsequently, the parameters were refined and the 3D structures were reconstructed using the FREALIGN package (*Grigorieff, 2007*) without imposing symmetry. The helical parameters were determined from the non-symmetrized map and the parameters were applied to the map. Twelve rounds of refinement with increasing resolution were performed. To assess the effective resolutions and possible over-fitting, the phases of high-resolution components (>10 Å) of individual images were randomized (*Chen et al., 2013*) and the 3D structures were reconstructed and analyzed by Fourier Shell Correlation (FSC; *Van Heel, 1987*) (*Figures 7* and *Figure 7—figure supplement 1*).

## Acknowledgements

We are grateful to N. Matsugaki for technical support. We thank members of the Hirokawa lab for assistance and discussions. We are grateful to J. Yajima for the professional suggestions on Microtubule gliding assay.

## Additional information

### Funding

| Funder | Grant reference number | Author |
|---|---|---|
| Ministry of Education, Culture, Sports, Science, and Technology | 23000013 | Nobutaka Hirokawa |
| Ministry of Education, Culture, Sports, Science, and Technology | 16H06372 | Nobutaka Hirokawa |
| Ministry of Education, Culture, Sports, Science, and Technology | 15K08168 | Ryo Nitta |
| Takeda Science Foundation | | Ryo Nitta |
| Japan Science and Technology Agency | | Masahide Kikkawa |

The funders had no role in study design, data collection and interpretation, or the decision to submit the work for publication.

### Author contributions

DW, RN, Conception and design, Acquisition of data, Analysis and interpretation of data, Drafting or revising the article, Contributed unpublished essential data or reagents; MM, Conception and design, Acquisition of data, Analysis and interpretation of data, Drafting or revising the article; HY, Conception and design, Analysis and interpretation of data, Drafting or revising the article, Contributed unpublished essential data or reagents; SI, Conception and design, Drafting or revising the article; HS, Acquisition of data, Analysis and interpretation of data, Drafting or revising the article; MK, Analysis and interpretation of data, Drafting or revising the article; NH, Conception and design, Analysis and interpretation of data, Drafting or revising the article

### Author ORCIDs

Ryo Nitta, http://orcid.org/0000-0002-6537-9272
Nobutaka Hirokawa, http://orcid.org/0000-0002-0081-5264

## Additional files

### Supplementary files

• Supplementary file 1. Crystal structure statistics for KIF19A motor domain 353WT. Data collection and refinement statistic table.

### Major datasets

The following datasets were generated:

| Author(s) | Year | Dataset title | Dataset URL | Database, license, and accessibility information |
|---|---|---|---|---|
| Wang D., Nitta R., Hirokawa N. | 2016 | Crystal Structure of the KIF19A Motor Domain Complexed with Mg-ADP | http://www.rcsb.org/pdb/explore/explore.do?structureId=5GSZ | Publicly available at the RCSB Protein Data Bank(Accession No. 5GSZ) |
| Morikawa M., Nitta R., Yajima H., Shigematsu H., Kikkawa M., Hirokawa N. | 2016 | Kinesin-8 motor, KIF19A, in the nucleotide-free state complexed with GDP-taxol microtubule | http://www.ebi.ac.uk/pdbe/entry/emdb/EMD-9538 | Publicly available at Electron Microscopy Data Bank (Accession No.9538) |
| Morikawa M., Nitta R., Yajima H., Shigematsu H., Kikkawa M., Hirokawa N. | 2016 | Kinesin-8 motor, KIF19A, in the nucleotide-free state complexed with GDP-taxol microtubule | http://www.rcsb.org/pdb/explore/explore.do?structureId=5GSY | Publicly available at the RCSB Protein Data Bank(Accession No. 5GSY) |

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
