## [Decision Letter]

Thank you for submitting your article "The Mechanism of Motility and Microtubule Depolymerization in Kinesin-8 motor, KIF19A" for consideration by *eLife*. Your article has been reviewed by four peer reviewers, and the evaluation has been overseen by a Reviewing Editor and Richard Aldrich as the Senior Editor. One of the four reviewers has agreed to reveal his identity: Robert Anthony Cross (Reviewer #2).

The reviewers have discussed the reviews with one another and the Reviewing Editor has drafted this decision to help you prepare a revised submission.

Summary:

This paper reports structure-function studies on mouse kinesin-8 KIF19A. Kinesins-8 combine the ability to move along microtubules with the ability to depolymerise the GTP caps of microtubules. Kif19A is a kinesin-8 that has been shown by the authors to be required for ciliary microtubule length-control. The structures reported here reveal novel features of the tubulin / microtubule binding surface of Kif19A. Wang et al. hypothesize that the various features of this surface allow Kif19A to bind to both 'straight' tubulin in microtubules and 'curved' tubulin in solution and at the depolymerising plus ends of dynamic microtubules. They test the roles of the various features of this interface by mutagenesis of KIF19A and by transplant experiments where these features are moved to related kinesins. The properties of these mutants are explored using depolymerisation assays done with GMPCPP microtubules, and in ATPase assays using both microtubules and tubulin. The depolymerisation mechanism(s) of kinesin-8 are poorly understood at the protein-chemistry level and KIF19A is particularly important, given its role in ciliogenesis. There is a previous structural study of a different kinesin-8 Kif18a, which reported some of the unusual features described here, but there is sufficient novel information that advances our understanding of the kinesin-8 mechanism.

Essential revisions:

1) The most notable aspect of the KIF19A structural findings is the extended helix a4 and its apparent retraction in the no NT cryoEM structure. However, the cryoEM data are at relatively low resolution, perhaps due to the fact that the occupancy by KIF19A is not saturated. Furthermore, interpretations in the Discussion of the differences between microtubule lattice and tip binding by KIF19A are neither clear nor convincing, given the poorly presented cryoEM data and absence of structural data for motor binding to curved microtubules. Density for KIF19A bound to microtubules without nucleotide is fit with KIF19A-ADP and nucleotide-free kinesin-1 crystal structures without clearly explaining this. The interpretation is that KIF19A helix a4 differs from the motor-ADP in the NT-free state and is similar to kinesin-1. However, the cryoEM density is not shown clearly enough in the Figure 6 images for the reader to evaluate this or other statements regarding KIF19A interactions with microtubules. This is one of the major findings of this paper and its presentation and description should be significantly improved.

2) Motility data for the truncated and mutant motors require further documentation. It should be indicated whether the leading or lagging microtubule ends were tracked and whether microscope drift was controlled for. Please also indicate the duration of assay/tracking times, the average microtubule length and whether microtubules were polarity marked. Please include kymographs of motility and explain whether the truncated monomer motor moves directionally.

3) For the PC2A mutant, please indicate whether microtubules were bound and not gliding, or not bound. If no microtubules were bound, were assays done with AMP.PNP to show sufficient motor attached to the surface to bind microtubules? Conclusions regarding the mutant should be formulated more carefully, based on the actual observations.

4) To determine which of the detected differences are significant, please use appropriate statistical tests throughout the paper.

5) The microtubule disassembly assays require better documentation and presentation: please state if assay is a disassembly assay, state that AMP.PNP is a control, provide time of incubation, state concentration of microtubule ends, define EC50 and state why this is used rather than Michaelis-Menten constant, state on plot axis that data are normalized, add error bars and indicate n=# to plots.

6) For the ATPase assays throughout, the authors' statements regarding effects are frequently not consistent with the data, e.g., they state no significant effects where data actually show significant but small effects, then follow with inaccurate conclusions regarding effects; they state no (or slight) effects where the data, in fact, show significant (or large) effects – these statements should be corrected. Number of independent assays (n=#) should be given in tables throughout.

7) Statements and interpretations throughout the manuscript frequently lack evidence or are inaccurate: e.g., processivity is not examined in this (or previous) studies – there is no evidence for the statements regarding KIF19A processivity in the subsections “Cryo-EM Reconstruction of 353WT Complexed with GDP-taxol MTs Revealed the Functional Anatomy of KIF19A”, “KIF19A Tethers Straight and Curved MTs Using Flexible Ionic Interactions” and”Model for KIF19A Motor Domain Function”. In the subsection “Asparagine 297 Contributes to the Adoption of MT and Tubulin Interfaces by Destabilizing Switch II Helices”, the proline is *highly* conserved, but not perfectly conserved among the kinesins. Please make the statements more accurate.

8) KIF19A is presumed to move on ciliary microtubules, which differ from the single microtubules used in the in vitro motility and kinetic assays reported here. The authors should take this into account in their interpretations, which are frequently overstated. For example, the velocities of KIF19A movement and depolymerization might be different on ciliary microtubules.

9) The coordinates, structure factors and EM electron density maps must be made available upon publication. Authors should provide statements regarding deposition of X-ray and cryoEM coordinates into public databases.

---

## [Author Response]

Essential revisions:

*1) The most notable aspect of the KIF19A structural findings is the extended helix a4 and its apparent retraction in the no NT cryoEM structure. However, the cryoEM data are at relatively low resolution, perhaps due to the fact that the occupancy by KIF19A is not saturated. Furthermore, interpretations in the Discussion of the differences between microtubule lattice and tip binding by KIF19A are neither clear nor convincing, given the poorly presented cryoEM data and absence of structural data for motor binding to curved microtubules. Density for KIF19A bound to microtubules without nucleotide is fit with KIF19A-ADP and nucleotide-free kinesin-1 crystal structures without clearly explaining this. The interpretation is that KIF19A helix a4 differs from the motor-ADP in the NT-free state and is similar to kinesin-1. However, the cryoEM density is not shown clearly enough in the Figure 7 images for the reader to evaluate this or other statements regarding KIF19A interactions with microtubules. This is one of the major findings of this paper and its presentation and description should be significantly improved.*

We thank the reviewer for this constructive comment. We have comprehensively revised Figure 7 and Figure 8 as well as the corresponding sections in the main text (subsection “Cryo-EM Reconstruction of 353WT Complexed with GDP-taxol MTs Reveals the Functional Anatomy of KIF19A” and the Discussion section). In the new figures, the atomic model of nucleotide-free KIF19A on a MT, produced using the rigid body fitted model of KIF19A-ADP and KIF5-free, is presented (in the second paragraph of the aforementioned subsection). This has enabled us to more clearly present the conformational changes from the KIF19A-ADP without MT to the KIF19A-free on an MT (Figure 7). In the Discussion section, in silico fitting of the KIF19A structures with a curved tubulin-dimer is described to clarify which regions are responsible for the dual functions of KIF19A (subsection “L8 and switch II of KIF19A requires flexibility to fit the tubulin interfaces in both the MT-lattice and MT-curved end”, first paragraph). This clearly illustrates that the KIF19A-specific conformations of the long L2, the retractable L8, and the disordered switch II, play central roles in its dual functions (Figure 8).

*2) Motility data for the truncated and mutant motors require further documentation. It should be indicated whether the leading or lagging microtubule ends were tracked and whether microscope drift was controlled for. Please also indicate the duration of assay/tracking times, the average microtubule length and whether microtubules were polarity marked. Please include kymographs of motility and explain whether the truncated monomer motor moves directionally.*

We have revised the text and figures as the reviewer suggests. Kymographs of motility tracking both the leading and lagging ends are presented in the revised manuscript. They clearly indicate the plus-end directed movement of the truncated monomer motor. A representative motility video of a truncated KIF19A monomer is also attached showing that most MTs move in a same direction but aggregate-like dots do not move directionally. This means that microscope drift is negligibly small.

*3) For the PC2A mutant, please indicate whether microtubules were bound and not gliding, or not bound. If no microtubules were bound, were assays done with AMP.PNP to show sufficient motor attached to the surface to bind microtubules? Conclusions regarding the mutant should be formulated more carefully, based on the actual observations.*

We apologize for the confusing presentation. In the revised manuscript, we have added the assay performed with PC2A in the presence of AMP-PNP. Although fewer MTs were observed compared with the WT motor, an increased concentration of PC2A (1 mg/ml) enabled sufficient microtubules to be bound (Figure 4—figure supplement 2). Even in these conditions, however, MTs were bound to the coverslip for too short a time to reliably calculate the gliding speed on the PC2A-coated coverslip. Thus, the PC2A mutation severely increased the dissociation constant for MTs so that few MTs were observed on PC2A attached to the coverslip. We have added this discussion in the first paragraph of the subsection “Loop L2 Tethers KIF19A to MTs and Might Slow Down Motility”.

*4) To determine which of the detected differences are significant, please use appropriate statistical tests throughout the paper.*

We have added appropriate statistical tests to determine significant differences.

*5) The microtubule disassembly assays require better documentation and presentation: please state if assay is a disassembly assay, state that AMP.PNP is a control, provide time of incubation, state concentration of microtubule ends, define EC50 and state why this is used rather than Michaelis-Menten constant, state on plot axis that data are normalized, add error bars and indicate n=# to plots.*

We are grateful for this constructive comment. We have revised the Methods (subsection “Microtubule Depolymerization Assay”), Figures (Figure 1, Figure 3, Figure 5 and Figure 6), and their legends as the reviewer suggests. In previous studies of KIF19A, as well as studies on MCAK/KIF2C, EC50 was used to characterize depolymerization ability (Hertzer et al., 2006; Niwa et al., 2012; Noda et al., 2012). To make a valid comparison with these previous reports, we used the EC50 value rather than the Michaelis-Menten constant.

*6) For the ATPase assays throughout, the authors' statements regarding effects are frequently not consistent with the data, e.g., they state no significant effects where data actually show significant but small effects, then follow with inaccurate conclusions regarding effects; they state no (or slight) effects where the data, in fact, show significant (or large) effects – these statements should be corrected. Number of independent assays (n=#) should be given in tables throughout.*

We have carefully checked and revised the manuscript to provide consistent interpretation for the ATPase assays. We have added the appropriate statistical tests to determine significant differences. We also added the statement of the number of independent assays in the figure legends.

*7) Statements and interpretations throughout the manuscript frequently lack evidence or are inaccurate: e.g., processivity is not examined in this (or previous) studies – there is no evidence for the statements regarding KIF19A processivity in the subsections “Cryo-EM Reconstruction of 353WT Complexed with GDP-taxol MTs Revealed the Functional Anatomy of KIF19A”, “KIF19A Tethers Straight and Curved MTs Using Flexible Ionic Interactions” and”Model for KIF19A Motor Domain Function”. In the subsection “Asparagine 297 Contributes to the Adoption of MT and Tubulin Interfaces by Destabilizing Switch II Helices”, the proline is* highly *conserved, but not perfectly conserved among the kinesins. Please make the statements more accurate.*

We have carefully revised the manuscript to make these statements more accurate.

*8) KIF19A is presumed to move on ciliary microtubules, which differ from the single microtubules used in the in vitro motility and kinetic assays reported here. The authors should take this into account in their interpretations, which are frequently overstated. For example, the velocities of KIF19A movement and depolymerization might be different on ciliary microtubules.*

We used single MTs to clarify the KIF19A-specific features compared with other kinesin motors. As the reviewer suggests, however, the velocities of KIF19A movement and depolymerization might be different on ciliary doublet microtubules. Thus we have carefully revised the manuscript to avoid over interpretation. We have also added a related statement at the end of the Discussion.

*9) The coordinates, structure factors and EM electron density maps must be made available upon publication. Authors should provide statements regarding deposition of X-ray and cryoEM coordinates into public databases.*

We have deposited all the structures solved here in PDB and EMDB. This statement has been added in the subsection “Additional Files”.